# The parietal cortex has a causal role in ambiguity computations in humans

**Gabriela Valdebenito-Oyarzo[1], María Paz Martínez-Molina[1], Patricia Soto-Icaza[1], Francisco Zamorano[2,3], Alejandra Figueroa-Vargas[1], Josefina Larraín-Valenzuela[1], Ximena Stecher[2], César Salinas[2], Julien Bastin[4], Antoni Valero-Cabré[5,6,7], Rafael Polania[8], Pablo Billeke [1] ***

**1** Laboratorio de Neurociencia Social y Neuromodulación, Centro de Investigación en Complejidad Social, (neuroCICS), Facultad de Gobierno, Universidad del Desarrollo, Santiago, Chile, **2** Unidad de Neuroimágenes Cuantitativas avanzadas (UNICA), Departamento de Imágenes, Clínica Alemana de Santiago, Santiago, Chile, **3** Facultad de Ciencias para el Cuidado de la Salud, Campus Los Leones, Universidad San Sebastián, Santiago, Chile, **4** Univ. Grenoble Alpes, Inserm, U1216, Grenoble Institut Neurosciences, Grenoble, France, **5** Causal Dynamics, Plasticity and Rehabilitation Group, FRONTLAB team, Institut du Cerveau et de la Moelle Epinière (ICM), CNRS UMR 7225, INSERM U 1127 and Sorbonne Université, Paris, France, **6** Cognitive Neuroscience and Information Technology Research Program, Open University of Catalonia (UOC), Barcelona, Spain, **7** Laboratory for Cerebral Dynamics Plasticity and Rehabilitation, Boston University, School of Medicine, Boston, Massachusetts, United States of America, **8** Decision Neuroscience Lab, Department of Health Sciences and Technology, ETH Zurich, Zurich, Switzerland

* pbilleke@udd.cl

**Data Availability Statement:** The complete minimal data set underlying the results used in our study are available in the public repository of the OSF website https://osf.io/zd3g7 under the name of

## Abstract

Humans often face the challenge of making decisions between ambiguous options. The level of ambiguity in decision-making has been linked to activity in the parietal cortex, but its exact computational role remains elusive. To test the hypothesis that the parietal cortex plays a causal role in computing ambiguous probabilities, we conducted consecutive fMRI and TMS-EEG studies. We found that participants assigned unknown probabilities to objective probabilities, elevating the uncertainty of their decisions. Parietal cortex activity correlated with the objective degree of ambiguity and with a process that underestimates the uncertainty during decision-making. Conversely, the midcingulate cortex (MCC) encodes prediction errors and increases its connectivity with the parietal cortex during outcome processing. Disruption of the parietal activity increased the uncertainty evaluation of the options, decreasing cingulate cortex oscillations during outcome evaluation and lateral frontal oscillations related to value ambiguous probability. These results provide evidence for a causal role of the parietal cortex in computing uncertainty during ambiguous decisions made by humans.

## Introduction

Making decisions in situations where the availability of outcomes is uncertain is a prevalent challenge in daily life. For example, when visiting a city for the first time, selecting a restaurant that serves the desired pizza can pose a dilemma. Making choices based on incomplete information about the availability of potential outcomes is a common phenomenon among humans

our manuscript — The parietal cortex has a causal role in ambiguity computations in humans (doi: 10.17605/OSF.IO/G2EA3). The additional toolbox and codes used in the analysis are available on our lab git site at https://github.com/neurocics/ and the OSF website https://osf.io/zd3g7.

**Funding:** This work was supported by Fondo Nacional de Desarrollo Científico y Tecnológico (FONDECYT, 1211227, 1181295 to PB, and 11230607 to PSI), Agencia Nacional de Investigación y Desarrollo (ANID, PAI77190047 to PSI, EQM150076 to PB, C12S03 to JB and PB). The funders had no role in study design, data collection, and analysis, decision to publish, or preparation of the manuscript.

**Competing interests:** The authors have declared that no competing interests exist.

**Abbreviations:** CTD, cluster threshold detection; DIC, deviance information criterion; FDR, false discovery rate; FEF, frontal eye field; HDI, high-density interval; ICA, independent component analysis; IPS, intraparietal sulcus; LOOIC, leave-one-out cross-validation information criterion; MCC, midcingulate cortex; MCMC, Markov Chain Monte Carlo; PDM, probabilistic decision-making; PPI, psychophysiological interaction; PPC, posterior parietal cortex; ROI, region of interest.

and other animals. These circumstances exhibit ambiguity, a type of uncertainty that may vary depending on the individual's knowledge of the environment [1]. Despite the common occurrence of ambiguous scenarios in ecological settings, organisms generally avoid such situations, a phenomenon known as ambiguity aversion [2–5]. Nevertheless, life often presents situations where ambiguity cannot be avoided, and individuals must make decisions based solely on available information. How humans make decisions when it is impossible to avoid ambiguity and what neurobiological mechanisms underlie such computations remain unclear.

When evaluating options and making decisions, individuals consider available information, such as the probability of outcomes and their associated rewards. This evaluation process appears to rely on a neural network consisting of the ventromedial prefrontal cortex, orbitofrontal cortex, and ventral striatum [6–8]. Moreover, the intraparietal sulcus (IPS) and dorsal posterior parietal cortex (PPC) [9–11] have been implicated in the perception and comparison of varying levels of uncertainty during decision-making. Additionally, research has shown a correlation between activity in the frontal and parietal areas and the degree of uncertainty and the updating process that reduces uncertainty [10,12]. Furthermore, recent studies in nonhuman primates have identified a crucial role for the parietal cortex in encoding the expected reduction of uncertainty during perceptual decision-making processes [13]. Activity in the parietal cortex has been associated with surprise signaling in human studies, but its connection to decision-making is unclear [14]. As a result, the specific role of the parietal cortex during decision-making under ambiguity and its causal contributions to these processes in humans remains elusive.

Evaluating the outcomes of our decisions is another essential aspect of decision-making behavior, as it enables us to learn and update our knowledge of the environment. Upon making a decision, individuals assess whether the outcome matches their expectations, resulting in a prediction error signal that can be detected in various brain regions, even in situations where no explicit learning occurs [15–18]. After a decision, research has revealed sustained connectivity between the parietal region and prefrontal cortex, which influences future decision-making [19,20]. Unexpected outcomes in uncertain situations have been linked to prefrontal activity, as indicated by fMRI studies [11] and oscillatory activity from EEG recordings [21,22]. A large body of work has shown a correlation between frontal delta and theta activity and prediction errors in uncertain situations [21,23–25]. Despite the well-established role of the connectivity between parietal and frontal regions in decision-making and value representations [20,26], the impact of the parietal activity on frontal prediction error signal arising from decisions under ambiguity is still unclear.

In this study, we aimed to investigate the hypothesis that the parietal cortex has a causal role in valuing ambiguous options when making decisions. Previous research has utilized the dichotomy between ambiguous and non-ambiguous options to analyze ambiguity [27]; however, this approach does not accurately depict real-life decision-making situations. To better understand the unique aspects of ambiguity computation in daily life decisions, we designed an experiment incorporating behavioral modeling and fMRI information to be tested in a subsequent TMS-EEG study. Our results showed that participants assigned some proportion of the unknown probability to objective, known probability during decision-making, increasing the uncertainty of their decisions. This process was linked to parietal activity during the decision period and midcingulate cortex (MCC) activity during feedback. Furthermore, TMS disruption of parietal activity led to an increase in unknown probability assignment and a reduction in delta activity in the MCC during feedback and theta activity during decision time, confirming a causal role of the parietal cortex in the computation of ambiguous information of the options during decision-making.

## Results

### Behavior

In the designed task, participants were required to choose between 2 options. Each option was associated with a different probability of being rewarded and a varying reward magnitude, as depicted in Fig 1. In half of the trials (ambiguity condition), the actual probabilities were partially concealed, resulting in varying degrees of ambiguity, ranging from 40% to 80% occlusion $P_a = \{0, 0.4, 0.5, 0.6, 0.7, 0.8\}$. This manipulation aimed to examine the effect of ambiguity on decision-making behavior. The probabilities and the rewards were misaligned, meaning that the option with the highest objective probability ($P_{obj}$) was not associated with the highest reward. This feature enabled us to calculate the rate at which participants chose the option with the highest probability despite offering a lower reward. Our results revealed that participants favored the option with the highest objective probability without ambiguity (no-ambiguity condition, rate = 0.65; Wilcoxon test, $n = 38$, $p = 3e\text{-}5$, logit mixed model, intercept: beta = 1.17, s.e. = 0.18, z-value = 6.1, $p = 6.8e\text{-}10$, d.f. = 1444; Fig 2A). This preference disappeared in the ambiguity condition (rate = 0.51, Wilcoxon test, $n = 38$, $p = 0.6$), resulting in a significant difference between the 2 conditions (Wilcoxon test, $n = 38$, $p = 8e\text{-}5$; mixed logit model, ambiguity dummy regressor: beta = −1.12, z-value = −7.1, $p = 8.4e\text{-}13$, d.f. = 1,444). This shift was linked to a negative correlation between choosing the highest probability and the level of ambiguity (rho = −0.25, $p = 0.0005$, d.f. = 187; mixed logit model, ambiguity weighted regression: beta = −3.2, s.e. = 0.6, z-value = −5.0, $p = 5\text{-}e7$, d.f. = 1,440; Fig 2B). Overall, participants shifted their preference from the greatest probability to the greatest reward in the ambiguity condition.

To determine whether the effect was due to changes in the probability or reward weight, we conducted logit models using the differences in probability and reward between options. The best-fit model indicated that, in no-ambiguity condition, probability and reward were considered (Logit mixed model: Probability regressor, beta = 12.8, s.e. = 1.5, z-value = 8.5, $p < 2e16$; Reward regressor, beta = 6.2, s.e. = 1.2, z-value = 5, $p = 4.7e\text{-}7$). However, ambiguity only impacted the weight of probability and not reward (Probability-ambiguity interaction, beta = −15.5, s.e. = 2.1, z-value = −7.1, $p = 1e\text{-}12$; Reward-ambiguity interaction, beta = −2.9, s.e. = 1.5, z-value = −1.8, $p = 0.06$). Fig 2D provides further detail using Logit hierarchical Bayesian estimation, with the posterior distribution of beta probabilities having a mean of 6.7 and 95% high-density interval (HDI) of [5.7–7.7] ($p_{MCMC} < 0.001$, $p_{MCMC}$ is a $p$-value derived by comparing the posterior distributions of the estimated parameters sampled via Markov Chain Monte Carlo (MCMC)). Similarly, the posterior distribution of beta rewards had a mean of 1.2 (HDI of [0.8–1.7], $p_{MCMC} < 0.001$). The posterior probability of beta Probability-Ambiguity interaction had a mean of −3.9 and HDI of [−4.6–3.1] ($p_{MCMC} < 0.001$). Meanwhile, the posterior beta Reward-Ambiguity interaction had a mean of −0.16 and 95% HDI of [−0.3 0.07] ($p_{MCMC} = 0.18$). In summary, the shift in behavior during ambiguous decisions appears to be associated with a selective decrease in probability weighting.

### Cognitive modeling of behavior

Next, we investigated the potential cognitive processes underlying the observed behavioral effects. Prior studies examining the choices between ambiguous and risky or safe options have employed 2 major modeling approaches. The first approach involves determining the degree of ambiguity aversion by adjusting a parameter that discounts the objective probability, assuming no decision bias (see [28,29] for detailed formulations). This discount parameter can take on positive values, indicating ambiguity aversion, or negative values, signifying ambiguity-

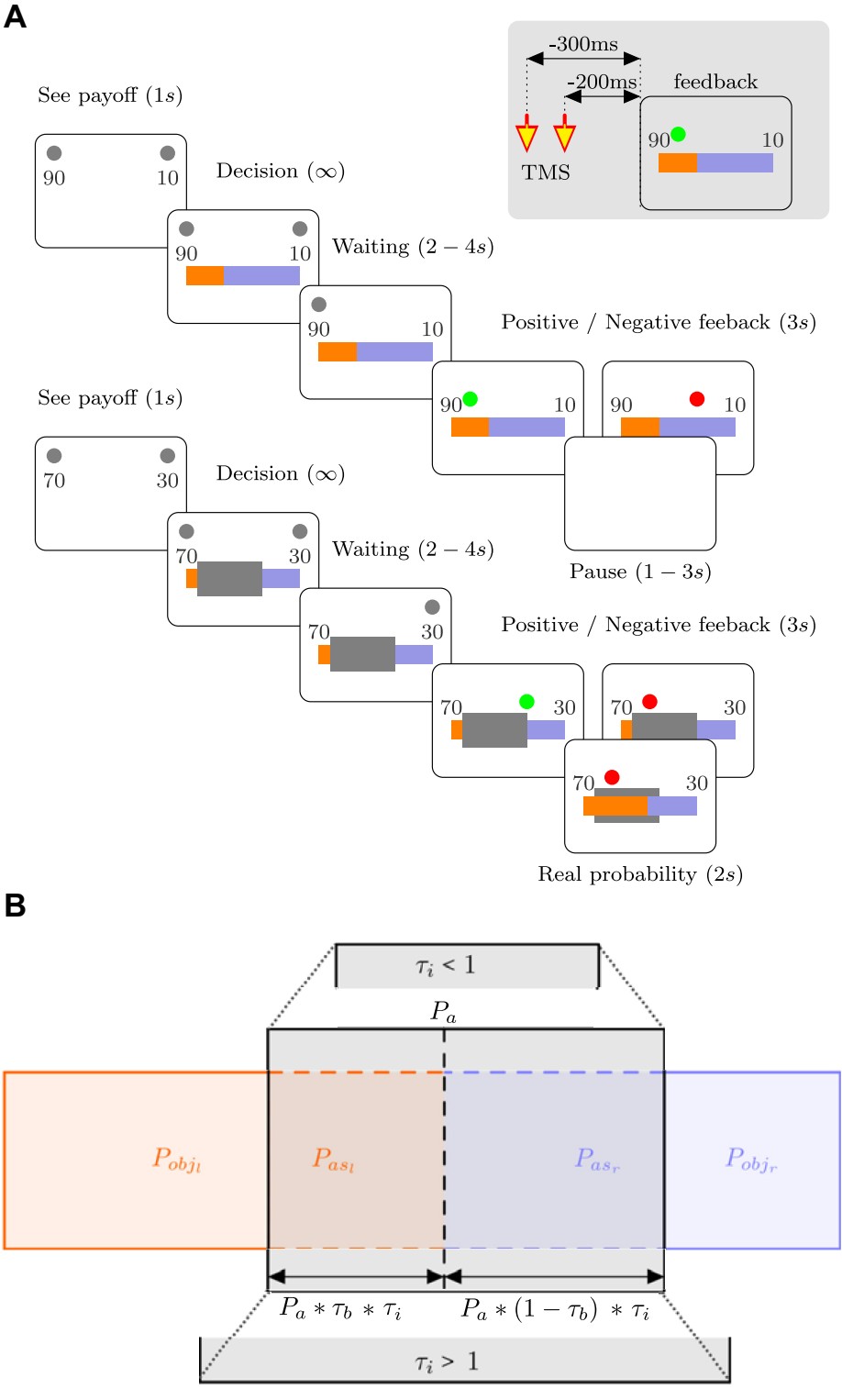

**Fig 1. Probability decision-making task. (A)** Task timeline. Participants were asked to decide between 2 options (left or right option). Each option had an associated reward indicated by a number. After a decision was made with a variable waiting time, feedback was provided. A green circle indicates that the participant won, whereas a red circle signals that he/she did not. In the ambiguity condition (bottom panel), a gray mask partially hides the color bars extension in the division. During the TMS-EEG session, a double TMS pulse is delivered −300 and −200 ms before feedback presentation, as represented in the gray rectangle over the superior right corner. **(B)** Schematic

representation of the objective probabilities ($P_{obj}$) and assigned probabilities ($P_{as}$), and the relationship among the ambiguity probability ($P_a$) and the model parameters, $\tau_i$ and $\tau_b$ (see the Results and Materials and methods sections for details).

seeking tendencies. This discount parameter, however, may not accurately capture the perceived degree of ambiguity or uncertainty in the decision-making process. The second approach involves parameters associated with attitudes toward ambiguity. These parameters capture an individual's optimism or pessimism regarding participants' beliefs about ambiguous information, e.g., [30]. This parameter can be interpreted as a bias parameter in assigning ambiguous information, assuming a constant perception of uncertainty. Based on the preceding formulations, we proposed a cognitive computation to distinguish between the perceived degree of ambiguity and potential probability assignment biases in the decision-making process.

In the ambiguous condition, we anticipated that participants would allocate a portion of the concealed area $P_a$ to the observable probability ($P_{obs}$). $\tau_i$ represents the portion of $P_a$ assigned to $P_{obj}$, and $\tau_b$ represents the laterality bias in this allocation (refer to Fig 1B). The probability that participants consider when making a decision, $P_{all}$, is determined by the following equations:

$$P_{all_l} = (P_{obj_l} + P_{as_l})/(P_{obj_l} + P_{as_l} + P_{obj_r} + P_{as_r}) \tag{1}$$

$$P_{as_l} = P_a \tau_b \tau_i \tag{2}$$

$$P_{as_r} = P_a(1 - \tau_b)\tau_i. \tag{3}$$

Subindices "l" and "r" represent the left and right options, respectively. A value close to zero for $\tau_i$ indicates a process in which participants do not assign unknown probabilities to objective (known) probabilities, ignoring the impact of the concealed area in the decision-making process. In other words, the uncertainty introduced by the concealed area is not considered in the decision-making process.

In contrast, when $\tau_i$ is greater, the hidden area is considered during the decision-making process, being assigned to objective probabilities. In the absence of bias between options, this results in a decrease in the difference between the options, increasing the uncertainty of the decision-making process. Values of $\tau_i$ greater than one indicate situations where participants make decisions as if they perceive more uncertainty than what is objectively generated by the hidden area (i.e., overestimating the uncertainty). On the other hand, values of $\tau_i$ less than one reflect situations where participants make decisions as if they perceive less uncertainty than what is objectively generated by the concealed area (i.e., underestimation of the uncertainty).

To test whether individuals perform such computations, we fitted several cognitive computational models based on prospect theory [30]. We used a hierarchical Bayesian approach [30,31], and model fits were evaluated using both the deviance information criterion (DIC) and the leave-one-out cross-validation information criterion (LOOIC, see Fig 2A). The model incorporating $\tau_i$ and excluding $\tau_b$ parameters showed the best data fit (see Fig 2A). The posterior distribution of the parameter $\tau_i$ was greater than 1, suggesting that participants overestimated the level of ambiguity ($\tau_i$ mean = 1.8; HDI = [1.2 2.4]; $p_{MCMC}$ = 0.001; Fig 2B and S1 Table). A correlation was observed between the $\tau_i$ value of each participant and the behavioral shift between ambiguity and no-ambiguity conditions (rho = −0.76, $p$ = 3e-7, df = 36; Linear regression correct for all the other model parameters, beta = −0.24, s.e. = 0.02, t-value =

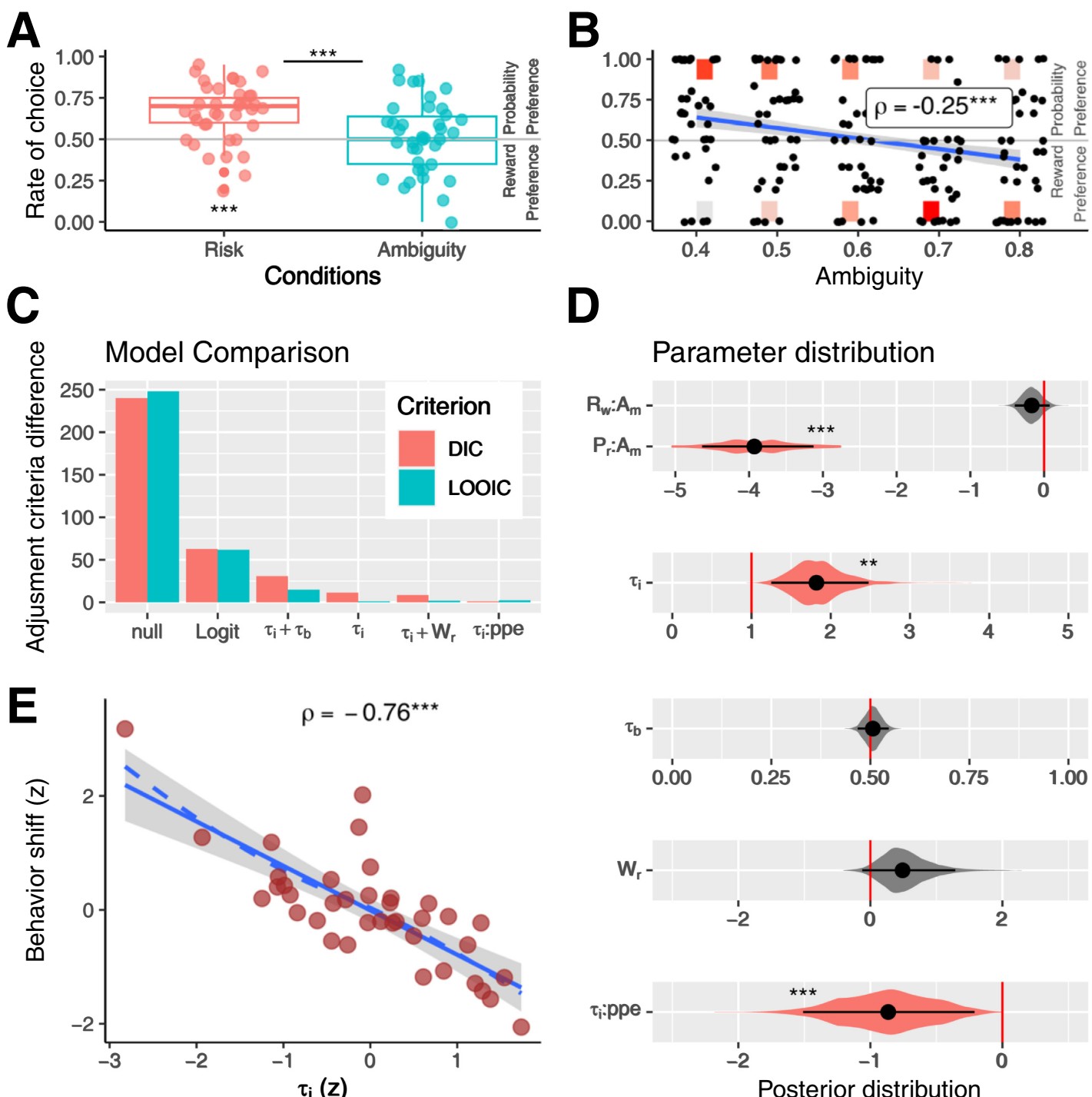

**Fig 2. Behavioral results.** (**A**) Rate of choice where individuals preferred the highest probability per conditions. (**B**) Rate of choice where subjects preferred the highest probability per degree of ambiguity. Black dots represent the rate per individual. Color rectangles indicate the number of individual decisions; red represents the maximum, and light gray represents the minimum account. The blue line represents the linear regression, and the gray area is the standard error. (**C**) Model fitting comparison using DIC, red, and LOOIC, green. (**D**) Posterior distribution of the key parameters for each model. Black dots represent the mean of the distribution, and black lines the 95% HDIs. The colored areas represent the complete posterior distribution. (**E**) Correlation between decision shift (difference between the rate of choices that the subject prefers the highest probability between condition, ambiguity less no-ambiguity conditions), and $\tau_i$ parameters. Red dots represent each subject. The solid blue line represents the linear regression, the dotted blue line the LOESS regression, and the gray area represents the standard error; * indicates $p < 0.05$, ** $p < 0.01$, ***$p < 0.001$. See also S1 Fig and S1 Table. The data underlying this figure can be found at https://osf.io/zd3g7/. DIC, deviance information criterion; HDI, high-density interval; LOOIC, leave-one-out cross-validation information criterion.

$-9.7$, $p$ = 4e-11, d.f. = 32; Fig 2E), demonstrating that this parameter reflects the behavioral change between conditions.

The posterior distribution of the parameter $\tau_b$ did not indicate a deviation from 0.5, suggesting there was no laterality bias in the ambiguity probability assignment ($\tau_i + \tau_b$ Model; $\tau_b$ mean = 0.5; HDI = [0.46 0.54]; $p_{MCMC}$ = 0.75; Fig 2B). We conducted 2 controlled analyses. One of these analyses incorporated a parameter ($W_r$, see Methods) to address potential biases towards options with greater rewards during ambiguous situations. The other analysis included 2 parameters ($\beta$ valency:$\tau_i$, $\beta$ ppe:$\tau_i$ that aimed to explore the possible influence of feedback valency (win or no win) and probability prediction error on the $\tau_i$ parameter in the subsequent trial. Although the results showed no significant adjustment improvement for the control models, these reveal 2 behavioral features. First, the $W_r$ parameter was no different from zero, indicating no evidence of ambiguity biasing decisions toward options with greater rewards ($W_r$ mean = 0.5; HDI = [−0.05 1.3]; $p_{MCMC}$ = 0.08; Fig 2D). Secondly, the value of $\beta$ ppe:$\tau_i$ was significantly less than zero, indicating that following unexpected feedback (e.i., with greater ppe), participants made adjustments in their subsequent decisions, leading to a reduction in the perceived uncertainty ($\beta$ ppe:$\tau_i$ mean = −0.87; HDI = [−1.5–0.21]; $p_{MCMC} <$ 0.001; Fig 2B). Nonetheless, feedback valency did not influence $\tau_i$ parameter ($\beta$ valency:$\tau_i$ mean = −0.1; HDI = [−0.31 0.09]; $p_{MCMC}$ = 0.3). Finally, we compared the adjustments made to the $\tau_i$ model with a discount parameter model (ambiguity aversions [28]). Our model demonstrated superior performance to this model (DIC difference = 19.5 and LOOIC difference = 50.2).

## Model simulations

We utilized the parameter readout from the $\tau_i$ model to simulate data using various generative $\tau_i$ values, replicating the same choice scenarios as those encountered by experimental participants. Our findings demonstrate the successful recovery of the generative $\tau_i$ values (see S2 Fig). Furthermore, we assessed how other variables vary as a function of $\tau_i$ (S3 Fig). Higher $\tau_i$ values were associated with a shift in preference towards more rewarding options, accompanied by an increase in the mean reward obtained. Consequently, this shift led to a reduction in the mean rate of winning, resulting in a greater mean reward prediction error. The probability prediction error exhibited a peak near $\tau_i$ around 1, declining for values both lower and higher than this threshold. Therefore, the influence of PPE in the $\tau_i$ of the next decision, as illustrated in Fig 2D, appears to be an effective means to diminish prediction errors in subsequent decision-making.

## Cognitive computation robustness

Finally, to verify the robustness of our proposed cognitive computation, we conducted a similar analysis on an independent sample of 20 participants. For this, the task was modified in several ways: the actual division of the mask was not revealed, the options were presented as 2 independent lotteries (while the underlying mechanism and instructions remained unchanged), and the participants made 150 decisions. The purpose was to investigate potential biases in ambiguity assignments. The results largely replicated the findings from the original sample (see S1 Fig). In the ambiguity condition, the participants shifted their decisions toward the option with the highest reward (Rate differences, Wilcoxon test, $n$ = 20, $p$ = 0.001), and the ambiguity degree was correlated with the decision shift (rho = −0.58, $p$ = 9e-6, d.f. = 48; mixed logit model, ambiguity weighted regression: beta = −3.3, s.e. = 0.5, z-value = −6.4, $p$ = 1e-10, d.f. = 2,991). The $\tau_i$ model outperformed other models, and individuals' $\tau_i$ values correlated with the decision shift (rho = −0.55, d.f. = 18, $p$ = 0.01). No evidence was found for laterality (mean

$\tau_b = 0.51$, HDI = [0.43 0.59], $p_{MCMC} = 0.6$) or reward (mean Wr = 0.11, HDI = [−0.2 0.4], $p_{MCMC} = 0.5$) biases and probability prediction error significantly reduce the $\tau_i$ in the next trials (mean β ppe:$\tau_i$ = −0.28, HDI = [−0.6–0.01], $p_{MCMC} = 0.04$). Unlike in the first sample, the posterior distribution of the $\tau_i$ parameter was no different from one, indicating that in these experimental conditions, the participants did not overestimate the uncertainty induced by the ambiguity condition (mean $\tau_i = 1.1$, HDI = [0.7 1.5], $p_{MCMC} = 0.6$).

## fMRI

**Value-related activity during decision-making.** We aimed to identify the brain regions involved in assigning probability in ambiguous situations. We used fMRI to measure brain activity during the decision phase and modeled the BOLD signal with various regressors (fMRI-Model 1: $R_w$, $P_{all}$, $P_a$, reaction time, and ambiguity condition; see Methods for more details). Results showed that the reward magnitude of the selected option ($R_w$) increased activity in regions such as the ventral striatum, while the degree of ambiguity ($P_a$) correlated mainly with bilateral IPS and PPC (Fig 3A). Then, to identify value-related areas, we investigated brain activity correlated with the probability of the chosen option during ambiguous decisions. For this, we first explored the contrast between ambiguity and no-ambiguity conditions in the correlation with $P_{all}$ (fMRI-Model 1, using $P_{all}$ regressors orthogonalized to $P_a$ to avoid confounding factors due to collinearity, and using subjects' $\tau_i$, see Methods). This analysis revealed a unique cluster in the right IPS that correlated with the probability of the chosen option during ambiguity (Fig 3A).

In order to uncover the specific role of parietal areas, we then used our cognitive model to identify particular ambiguity computation. As $\tau_i$ reflects how the subject incorporates the uncertainty derived from the concealed area in the decision-making process, we used fixed $\tau_i$ to calculate $P_{all}$ as if subjects underestimate uncertainty ($\tau_i = 0$) compared with objective uncertainty ($\tau_i = 1$) in the decision-making process. To do it, we tested an fMRI model that includes both regressors $P_{all(\tau i = 0)}$ and $P_{all(\tau i = 1)}$, orthogonalizing between them to identify the specific participation of brain areas in those proceeding (fMRI-Model 2, see Methods). The results showed that the right parietal areas, including the IPS and the PCC, were uniquely correlated with $P_{all(\tau i = 0)}$, suggesting that these regions are involved in a computation that underestimates the uncertainty of the options. In contrast, left parietal regions correlated with both $P_{all(\tau i = 0)}$ and $P_{all(\tau i = 1)}$, and $P_{all(\tau i = 1)}$ also correlated with other brain areas, including the MCC and the striatum (see S2 Table). These findings indicate that the parietal regions play a role in processing ambiguity and adjusting the level of uncertainty during decision-making. Specifically, the right parietal regions tend to compute options as more certain than they actually are.

**Feedback-related activity.** As we identified that probability prediction error influences the following decision, we investigated brain activity related to feedback processing, looking for specific prediction error-related activity. For this, we employed an fMRI model with multiple regressors to distinguish activity related to the probability prediction error (unsigned prediction error, uPE-$P_{all}$) and the reward prediction error (uPE-$R_w$, see Methods). Results showed that winning (positive feedback) elicited activity in several brain regions, with a peak in the bilateral ventral striatum (Fig 3B). For ambiguity and no-ambiguity decisions, uPE-$P_{all}$ showed correlated activity in the MCC with no significant difference between the conditions. No significant modulation was observed for uPE-$R_w$.

As the parietal cortex is implicated in neural computations regarding decision-making uncertainty, it should additionally influence the formation of outcome expectations and prediction errors. Hence, we expect to observe increased connectivity between this region and

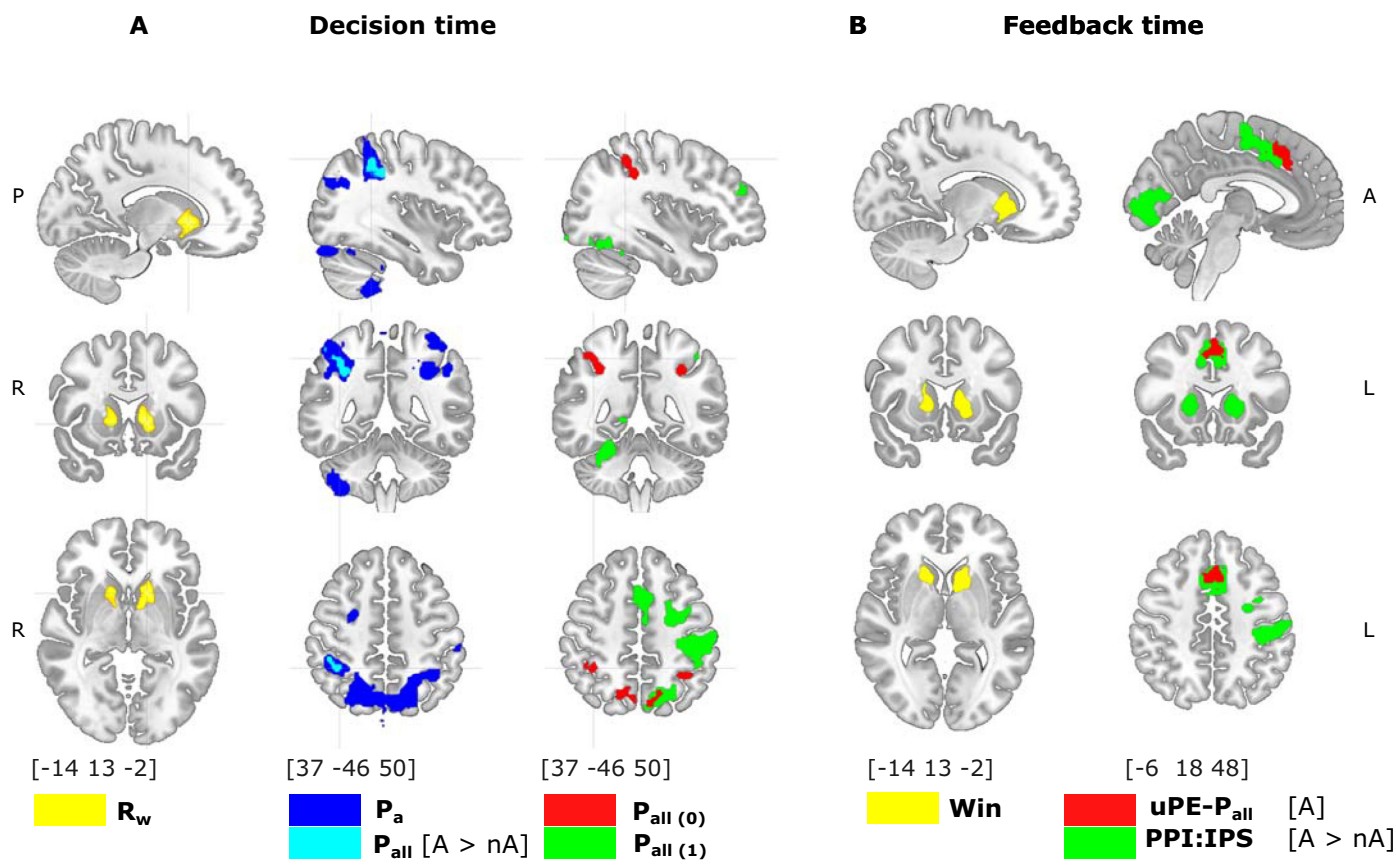

**Fig 3. Brain activity during decision-making and feedback.** (**A**) Brain activity during decision-making. The reward magnitude ($R_W$, yellow) of the chosen option is related to the activity in the ventral striatum (CTD Z = 3.1, cluster corrected $p$-value < 3e-8, for visualization threshold Z = 4). The degree of ambiguity ($P_a$, blue) correlates with the IPS and the PCC (cluster corrected $p$-value < 1e-10), among other areas. The probability assigned during ambiguity correlated with the right IPS (light blue, $P_{all}$[A > nA]:contrast $P_{all}$ during ambiguity > $P_{all}$ during no-ambiguity condition, corrected $p$-value = 0.0002). Underestimating uncertainty (red, $P_{all(\tau_i = 0)}$ calculated with $\tau_i = 0$) correlated bilaterally with the IPS (corrected $p$-value = 0.0003) and the PCC (corrected $p$-value < 1e-10). Objective uncertainty (green, $P_{all(\tau_i = 1)}$ calculated with $\tau_i = 1$) correlated with the IPS, the PCC, the somatosensory area in the left hemisphere, and the SMA (all corrected $p$-value < 1e-10). (**B**) Brain activity during feedback. The fact of winning (Win, yellow) correlated with ventral striatum activity (CTD Z = 3.1, cluster corrected $p$-value < 1e-5, for visualization threshold Z = 4). The probability prediction during ambiguity (uPE-$P_{all}$[A], red) correlated with activity in the MCC (CTD Z = 3.1, cluster corrected $p$-value < 1e-5). Contextual brain connectivity (PPI) (IPS seed from $P_{all}$[A > nA] contrast) showed that the ambiguity condition generates an increase in the correlation between the IPS and several brain regions, including the MCC and the ventral striatum. See also S2 Table. The data underlying this figure can be found at https://osf.io/zd3g7/. CTD, cluster threshold detection; IPS, intraparietal sulcus; MCC, midcingulate cortex; PPI, psychophysiological interaction.

areas related to outcome monitoring during feedback. Thus, we conducted a connectivity analysis (psychophysiological interaction, PPI, analysis) and tested increases in contextual connectivity of IPS (seed in joint activity for $P_{all}$ [A > nA] and $P_{all(\tau_i = 0)}$) during ambiguity (see Methods). This region increased connectivity with several outcome-monitoring areas, including the MCC and the striatum (Fig 3B).

**Predicting behavior.** To assess the influence of these identified areas on behavior, we examined whether BOLD activity in the parietal region could predict behavioral shifts between non-ambiguous and ambiguous conditions, as observed in the behavioral analysis (Fig 2A). We employed mixed logit models to predict whether subjects preferred the more probable or rewarded option on a trial-by-trial basis. We extracted the BOLD signal during the decision-making period from a region of interest (ROI), identified by the $\tau_i = 0$ model (fMRI-Model 2) and an MCC ROI based on uPE-Pall [A] analysis (see Fig 3). Consistent with behavioral

analysis, participants in the ambiguity condition shifted their preference toward the more rewarded option (ambiguity regressor: beta = −1.08, s.e. = 0.1, t = −8.8, $p < 1e$-10). Parietal activity significantly attenuated this effect (Parietal-ambiguity interaction: beta = 0.27, s.e. = 0.12, t = 2.2, $p = 0.02$, d.f. = 1,436) in line with the interpretation that the parietal cortex reflects decisions made with greater certainty (see Fig 3A). No impact was observed during the no-ambiguity condition (Parietal regressor: beta = −0.15, s.e. = 0.1, t = −1.5, $p = 0.12$;). No significant effects were found for the MCC ROI (MCC-Ambiguity interaction: beta = 0.15, s.e. = 0.11, t = 1.2, $p = 0.19$, d.f. = 1436). In the parietal model, the inclusion of the interaction between parietal signal and ambiguity resulted in a significant improvement in the fit (Chi-squared = 4.9, $p$-value = 0.02). In contrast, for the MCC model, there was no such improvement (Chi-squared = 1.6, $p$-value = 0.1947). However, we did not find evidence of a fit difference between the parietal and MCC models (Chi-squared = 1.9, $p$-value = 0.16).

## TMS-EEG

**Parietal inhibition increased the assignment of ambiguous probability.** To investigate the causal role of the parietal cortex in ambiguous probability assignments, we conducted a TMS-EEG experiment while participants performed the same task as in the fMRI experiment. Expecting a similar behavioral effect, we targeted 2 regions in the parietal cortex that showed a correlation with $P_a$ and $P_{all(\tau i = 0)}$ but not with $P_{all(\tau i = 1)}$ (as shown in Fig 4A). These regions were the dorsal PPC (MNI: [14, −64, 56]) and the right intraparietal sulcus (IPS, MNI: [46, −44, 57]). Vertex stimulation was used as a control condition. Since we found contextual connectivity between parietal and frontal regions during feedback and a behavioral effect of prediction error in the subsequent decision, we designed an online TMS stimulation consisting of a doublet of pulses (separated by 100 ms, hence covering a time window of approximately 100 to 200 ms) delivered trial-by-trial following the decision-making process (200 ms before the feedback onset). The rationale behind those choices was to disrupt the outcome expectancy derived from ambiguity computation, which is required to encode the prediction error during feedback. Utilizing this protocol, we avoided interfering with the decision time in which the TMS pulse could cause motor and attentional biases. Furthermore, this approach allowed us to discern behavior effects arising from the cumulative impact of TMS stimulation. This cumulative effect could be expected by the sustained perturbation of the probability prediction error signal, which in turn influences the $\tau_i$ parameter across successive trials, as illustrated by the behavioral analyses shown in Fig 2D ($\tau_i$:ppe interaction).

Following the above consideration, we first aimed to examine the intra-block cumulative effect of trial-by-trial TMS on behavior. To do this, we analyzed the behavioral effects in the last 20 trials of each 40-trial block of the same TMS stimulation (see Methods). Considering that the targeted regions are linked to the underestimation of uncertainty in the decision-making process (as evidenced by the correlation with $P_{all(\tau i = 0)}$ and the reduced behavioral shifts in response to ambiguity), we hypothesized that inhibiting these areas would amplify the behavior effects of ambiguity. Consequently, both vertex and parietal TMS stimulation showed significant decision shifts between the no-ambiguity and ambiguity conditions (rate differences Vertex TMS = −0.11, confidence interval = [−0.19–0.02]; Wilcoxon test $p = 0.02$; Parietal TMS = −0.2, confidence interval = [−0.25–0.14]; $p = 4e$-5), with parietal TMS stimulation having a greater effect (mean difference = 0.09, confidence interval = [0.02 0.15], Wilcoxon test, $p = 0.01$). Parietal stimulation also increased the correlation between ambiguity degree and decision shift (linear mixed model, ambiguity degree and TMS interaction, beta = −0.7, s.e. = 0.3, z-value = −2.2, $p = 0.02$, d.f. = 2,737). There was no evidence of a difference between IPS and PPC stimulation (linear mixed model, PPC-IPS difference, beta = 0.6, s.e. = 0.43, z-

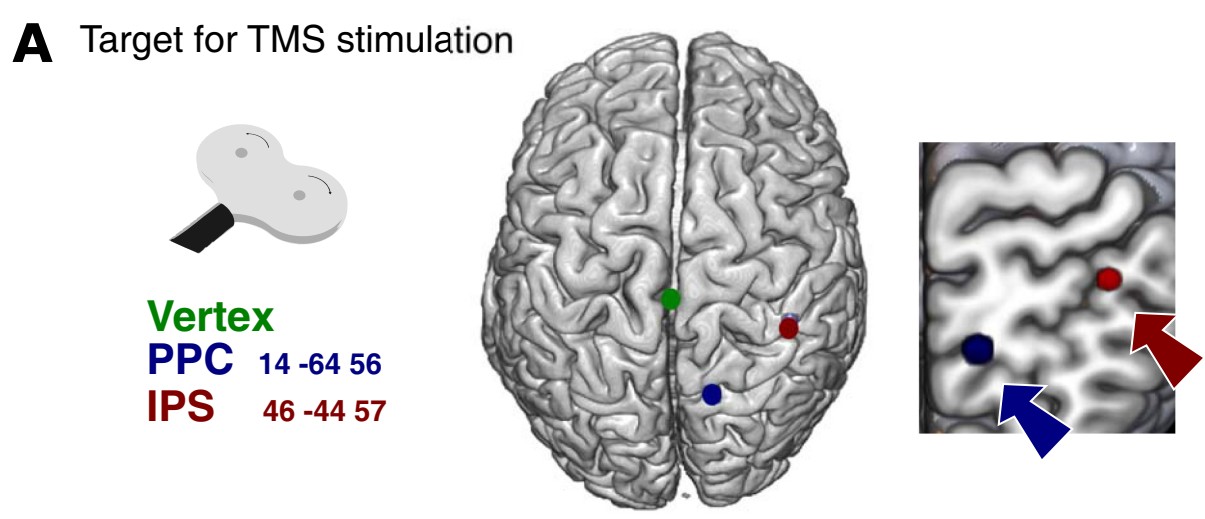

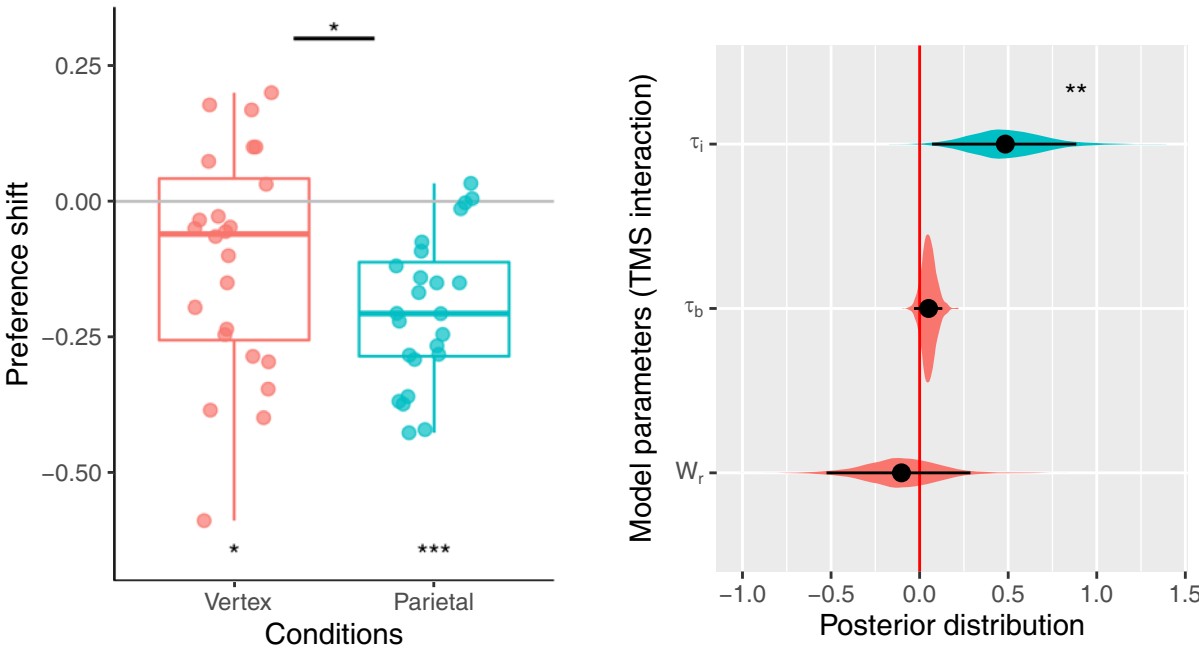

**Fig 4. Behavioral result of interleaved EEG-TMS experiments.** (**A**) Target areas for TMS stimulation (right PPC x = 14, y = −64, z = 56; right IPS x = 46, y = −44, z = 57, and Scalp Vertex). (**B**) Decision shift (difference between the rate of choices subjects prefer with the highest probability between conditions, no-ambiguity less ambiguity) comparison between Vertex and Parietal TMS stimulation. (**C**) Posterior distribution effect of TMS stimulation on key parameters for cognitive models. Black dots represent the mean of the distribution, and black lines represent the 95% HDIs. The colored areas represent the complete posterior distribution. The data underlying this figure can be found at https://osf.io/zd3g7/. HDI, high-density interval; IPS, intraparietal sulcus; PPC, posterior parietal cortex.

value = 1.4, $p$ = 0.14, d.f. = 2,722). Accordingly, the separated analysis for both TMS stimulation showed that both significantly modulated behavior (linear mixed models, ambiguity and IPS interaction, beta = −0.32, s.e. = 0.15, z-value = −2.0, $p$ = 0.03, d.f. = 2,737, ambiguity and PPC interaction, beta = −0.39, s.e. = 0.15, z-value = −2.6, $p$ = 0.009, d.f. = 2,737; ambiguity degree and IPS interaction, beta = −0.51, s.e. = 0.24, z-value = −2.1, $p$ = 0.03, d.f. = 2,737,

ambiguity degree and PPC interaction, beta = −0.59, s.e. = 0.23, z-value = −2.4, $p = 0.01$, d.f. = 2,737).

The logit model shows that parietal TMS stimulation increases the impact of the ambiguity condition on probability weighting without affecting reward weighting. The interaction between probability, ambiguity, and TMS was significant (beta = −4.86, s.e. = 2.3, z-value = −2.1, $p = 0.03$, d.f. = 2,788); meanwhile, the interaction between reward, ambiguity, and TMS was not significant (beta = −0.2, s.e. = 2.2, z-value = 0.08, $p = 0.9$). Logit hierarchical Bayesian estimation showed the same effects (posterior distribution of beta for probability, ambiguity, and TMS interaction: mean = −3.3, HDI = [−7.4–0.1], $p_{MCMC} = 0.027$, and posterior distribution of beta for reward, ambiguity, and TMS interaction: mean = 0.3, HDI = [−3.1 4.2], $p_{MCMC} = 0.4$). Subsequently, we assessed whether the behavioral changes induced by TMS stimulation had any discernible impact on the overall reward acquired by participants. Our analysis revealed no significant alterations in reward levels or the accumulation of positive feedback (mixed model, Reward: TMS regressor beta = −0.12, s.e. = 0.2, z-value = −0.5, $p = 0.6$; TMS-ambiguity interaction beta = 0.15, s.e. = 0.2, z-value = 0.5, $p = 0.6$; Positive feedback (win): TMS regressor beta = 0.03, s.e. = 0.1, z-value = 0.2, $p = 0.8$; TMS-ambiguity interaction beta = 0.003, s.e. = 0.1, z-value = −0.01, $p = 0.9$).

**Cognitive computational modeling for TMS stimulation.**   Additionally, we investigated the impact of TMS stimulation on the parameters of cognitive models. Our results showed that the $\tau_i$ model performed better than the other models and that TMS stimulation increased the $\tau_i$ parameter (posterior distribution of interaction $\tau_i$ and TMS, mean = 0.66, HDI = [0.07 0.88], $p_{MCMC} = 0.008$, see Fig 4), and that none of the other model parameters were affected by TMS stimulation (see S3 Table and below). We also examined the interaction between parameters that reflect lateral bias to assess a potential attention bias generated by right parietal inhibition [32]. None of these showed significant TMS modulation ($\beta_0$ of softmax and TMS interaction mean = −0.04, HDI = [−0.2 0.1], $p_{MCMC} = 0.6$; $\tau_b$ and TMS interaction mean = 0.17, HDI = [−0.03 0.12], $p_{MCMC} = 0.17$). Finally, we did not find evidence for modulation in reward estimation due to TMS stimulation ($\alpha$ parameter of prospect theory and TMS interaction mean = 1.6, HDI [−3.8 8.4], $p_{MCMC} = 0.5$; $W_r$ and TMS interaction mean −0.1, HDI = [−0.5 0.2], $p_{MCMC} = 0.6$). These findings suggest that the disruption of parietal activity specifically influences the assignment of ambiguous probabilities during decision-making processing without affecting other computations. We subsequently investigated potential differences between IPS and PPC stimulation effects. Both stimulations independently led to an increase in the $\tau_i$ parameter, with no statistically significant difference observed between them (posterior distribution of interaction between $\tau_i$ and TMSppc, mean = 0.62, HDI = [0.06 1.27], $p_{MCMC} = 0.012$; $\tau_i$ and TMSips interaction, mean = 0.33, HDI = [0.01 0.72], $p_{MCMC} = 0.045$; Difference TMSppc—TMSips mean = 0.28, HDI = [−0.29 0.97], $p_{MCMC} = 0.35$). Finally, we tested whether the TMS stimulation changed behavior, modulating the influence of feedback valency of PPE in a subsequent trial. This analysis reveals no specific modulation due to TMS stimulation in these parameters (TMS:ppe:$\tau_i$ mean = −0.3, HDI = [−3.2 1.6], $p_{MCMC} = 0.5$; TMS:valency:$\tau_i$ mean = 0.9, HDI = [−1.6 10.9], $p_{MCMC} = 0.8$).

**Parietal inhibition interrupts the prediction error signal related to ambiguous probability.**   We assessed the impact of parietal inhibition on the EEG oscillatory activity during feedback. TMS stimulation showed that subjects acted as if their decisions were more uncertain (overestimation of the uncertainty). Hence, we specifically examined the signal linked to uncertainty using a model with varying $\tau_i$, similar to the fMRI model 2 (see Methods). We anticipated that the prediction error signal of the probability calculated with $\tau_i = 0$, representing a lower degree of uncertainty in decision-making, would be particularly affected by parietal

TMS stimulation. For this, we explored frontal electrodes where oscillatory activity related to prediction error has been described in prior work [21–23,33].

The results showed that after feedback, an oscillatory activity in the delta range with a peak frequency of [2.3 3.6] Hz (Fig 5A) was generated by probability prediction error. Orthogonalizing uPE-P$_{all(\tau i = 0)}$ to uPE-P$_{all(\tau i = 1)}$ regressors, results showed that the delta activity reflected a distinct activity related to the ambiguity decision process, associated with decisions made underestimating uncertainty (cluster [1.5 4] Hz, [0.2 0.75] seconds post feedback, as determined by a cluster-based permutation test in frontal electrodes, cluster threshold detection (CTD) $p < 0.05$, corrected $p = 0.008$). As expected, parietal TMS stimulation disrupted this signal, resulting in a negative effect in the delta range (cluster [1.8 4] Hz, [0.3 0.78] seconds post-feedback, cluster-based permutation test in frontal electrodes, CTD $p < 0.05$, corrected $p = 0.03$). Source analysis revealed that the modulation caused by parietal TMS stimulation involved a similar area as found in the fMRI experiment in the MCC (as shown in the inserts in Fig 5B). The separate analysis of the IPS and PPC TMS stimulations revealed that both modulated the MCC delta activity to varying degrees (Fig 5C). When comparing these TMS stimulations, a modulation effect was observed near the PPC stimulation site (see Fig 5C).

**Parietal inhibition interrupts the value signal related to ambiguous probability.** As TMS generated behavioral effects, we expected changes in brain activity related to decision time. Thus, we investigated the modulation of oscillatory features during the decision period induced by TMS stimulation. We focused on the time interval preceding participants' button presses. Similar to our fMRI analysis, we conducted correlations between brain signals and multiple regressors, including the probability of the chosen option, while contrasting the Ambiguity and No-ambiguity conditions (Pall[A > nA]) within the last 20 trials per block where behavioral effects were observed. Since we did not have specific hypotheses regarding the oscillatory features and electrode locations implicated in this modulation, we conducted an exploratory whole-scalp analysis. We identified 2 clusters within the theta and alpha frequency ranges, localized in frontal and central electrodes (Fig 6A, cluster-based permutation test corrected $p < 0.05$). Source estimation of the theta modulation ([4–8] Hz and [−0.9 to −0.4] s) revealed its proximity to the frontal eye field (FEF), a region that had shown a correlation with the degree of ambiguity (P$_a$) in the fMRI analysis (Fig 6A, magenta line). Parietal TMS inhibition demonstrated a negative modulation of this activity in the frontocentral electrode and the FEF region in the source space (Fig 6B). The separate analysis of IPS and PPC TMS stimulations revealed that only IPS exhibited significant modulation in the FEF region, with no differences between the 2 stimulations (Fig 6C).

Overall, the EEG results suggest that disrupting parietal activity before feedback impacts the oscillatory activity in the MCC evoked by the prediction error and in the FEF during decision-making, suggesting a decision and feedback processing computed as if the uncertainty of the options were overestimated. Thus, the parietal cortex plays a causal role in ambiguity computation, and parietal-frontal interaction is necessary for signaling the outcome predictions made through this ambiguity computation.

## Discussion

The results of this study provide evidence for a causal role of the parietal cortex in decision-making under ambiguous conditions. Using consecutive analyses and sequentially informed fMRI and EEG-TMS experiments, we investigated the cognitive processes involved in decision-making in ambiguous situations and tested the causal involvement of the parietal cortex. We found that the subjects incorporated the uncertainty from the ambiguous information by shifting their preferences. This behavior can be explained by a model that evaluates how

# Prediction error related activity

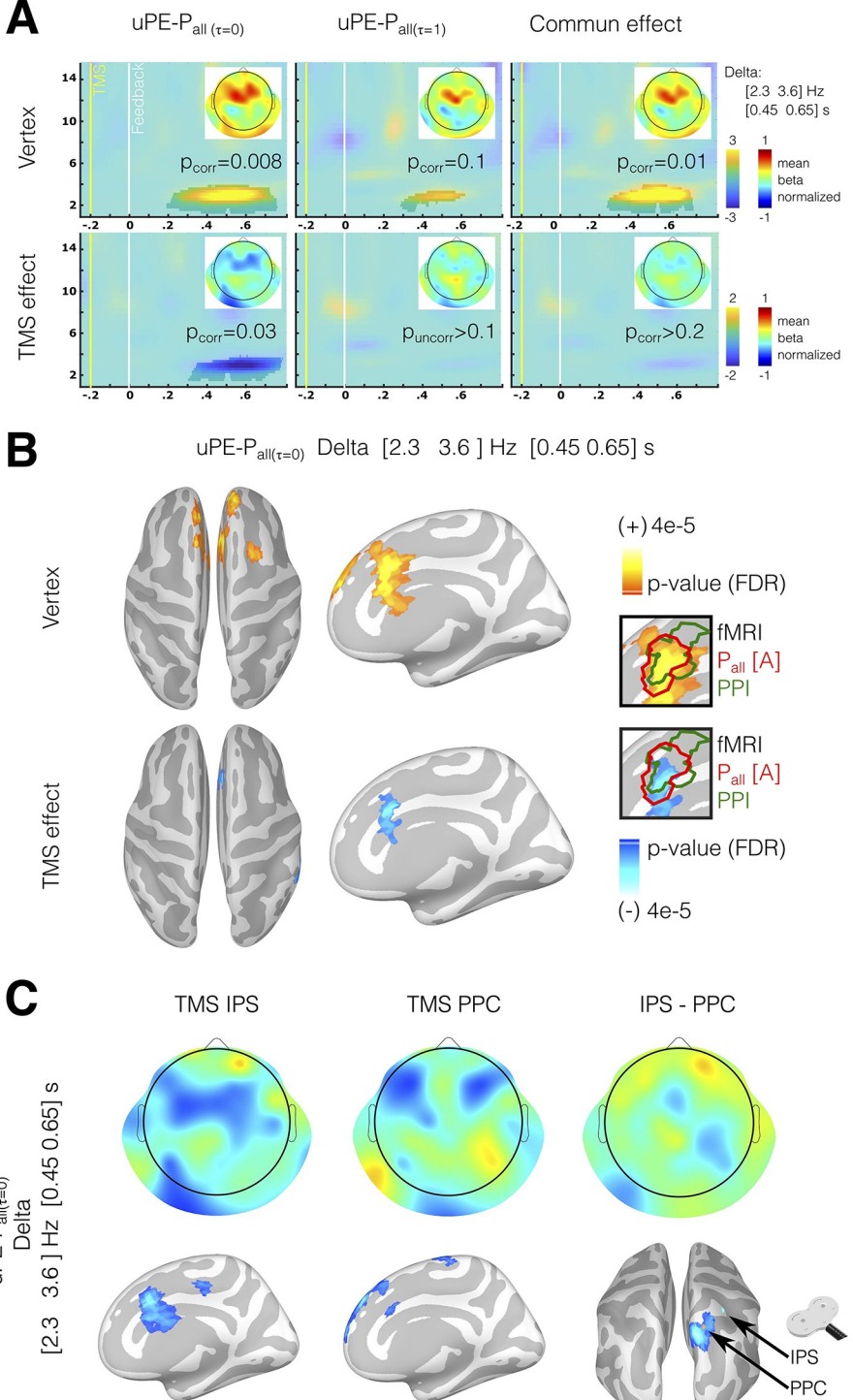

**Fig 5. Oscillatory brain activity in frontal electrodes associated with unsigned prediction error during feedback.**
(A) Time-frequency chart in frontal electrodes for the correlation between oscillatory power and unsigned prediction error is given by $\tau_i = 0$ model (uPE-$P_{\text{all}(\tau_i = 0)}$), $\tau_i = 1$ (uPE-$P_{\text{all}(\tau_i = 1)}$) model, and de join effect of 2 models for both Vertex TMS stimulation and the difference between vertex TMS and parietal TMS stimulation (TMS effect). The highlighted areas indicate time-frequency epochs showing significant modulation (without time-frequency a priori,

whole-scalp cluster-based permutation test, CTD: $p < 0.05$ Wilcoxon test). Scalp topographies show oscillatory activity in the delta range. (B) Source estimation for delta activity correlated with unsigned prediction error given by $\tau_i = 0$ model (uPE-P$_{all(\tau i = 0)}$) for Vertex and TMS effect. Sources that survive multiple comparison corrections are shown (FDR q < 0.05). The highlighted areas (green and red lines in the inserts) represent the coincident areas for EEG source estimation and BOLD activity for the fMRI experiment. All source results are shown in a high-resolution mesh only for visualization purposes. (C) A separate analysis of delta activity for TMS stimulation in the IPS and PPC and the differences between them. The data underlying this figure can be found at https://osf.io/zd3g7/. CTD, cluster threshold detection; FDR, false discovery rate; IPS, intraparietal sulcus; PPC, posterior parietal cortex.

individuals incorporate environmental uncertainty into their decisions. Neurobiological analyses show that bilateral parietal activity was linked to objective uncertainty, while right parietal activity was linked to a process that led to incorporating less uncertainty during decision-making. Indeed, such a process is specifically affected by the interference of the right parietal activity evoked by time-locked TMS perturbation. Inhibition of the right parietal cortex generated that individuals behave as they perceive more uncertainty in the decision processing, decreasing the frontal oscillatory activity related to both the value process during a decision and prediction error during feedback.

Beyond the known role of the parietal region in perceptual decision-making [34], increasing evidence has related parietal activity to value during decision-making under conditions of uncertainty [1,35]. Nonhuman primate studies have shown that parietal regions, such as the IPS, link the probability of obtaining a reward with a specific action (e.g., the direction of the saccade [36]). Neurons of the dorsal parietal region have also shown activity for a combination of reward magnitude and probability [1]. Moreover, some parietal neurons are specifically modulated by the expected utility of the options [37]. Following this notion, research comparing human decision-making models has shown a selectivity of the parietal cortex in encoding expected utility (i.e., the weight of the reward given by the subjective probability as expressed in Prospect Theory [35]). In this context, our results show differential modulations of the parietal cortex associated with the chosen option probability depending on the objective and perceived uncertainty. Research on nonhuman primates suggests that parietal neurons are highly attuned to the level of uncertainty in a perceptual decision-making task [13]. In addition, these findings have highlighted the critical role played by the parietal cortex in encoding information about the potential reduction of uncertainty resulting from a particular behavior or decision [13]. Taken together, these studies suggest that the parietal cortex is able to gauge the predictability of reward by distinguishing between what is already known and what remains uncertain.

Our experimental approach found a causal role of the parietal cortex in the computation of ambiguous options, even though human studies have shown conflicting results on its role in decision-making under uncertainty. Following our results, patients with posterior parietal lesions are less able to adjust their decision-making strategies based on the probability of winning than patients with frontal lesions, suggesting a potential causal role of the PPC in decision-making [38]. On the other hand, parietal activity has been related to a surprise signal with a general effect of cognitive reallocation, for example, slowing reaction time, but not with a value process [14]. In addition, other studies have reported a correlation between parietal activity and value processing only in specific demanding circumstances, for example, time pressure [39]. Parietal activity has also been correlated with the belief update, reducing the degree of ambiguity rather than value update [12]. In this context, our results support a causal engagement of the parietal cortex in value processing under uncertain situations, as parietal interference by TMS affected a particular computation related to ambiguity management. Additionally, parietal suppression reduced the value and prediction error signals associated

# Decision related activity

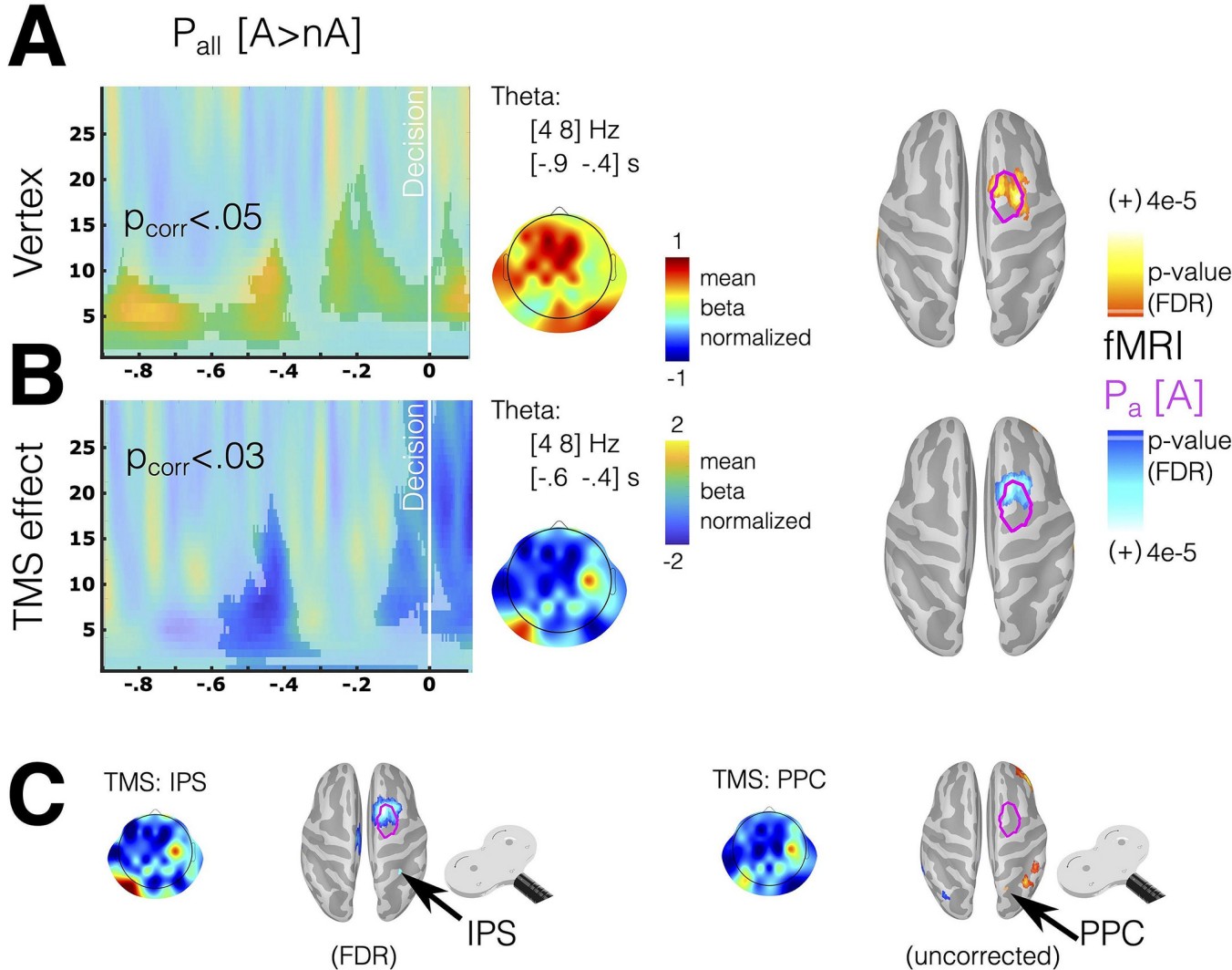

**Fig 6. Oscillatory brain activity in frontal electrodes associated with the probability of the chosen option during the decision period.** (A) The time-frequency chart illustrates differences in frontal electrodes between the Ambiguity and No-ambiguity conditions in the correlation between oscillatory power and the probability of the chosen option (Pall[A > nA]). (B) The joint effect of 2 models, one for Vertex TMS stimulation and the other for the difference between Vertex TMS and parietal TMS stimulation (TMS effect). (C) A separate analysis of theta activity for TMS stimulation in IPS and PPC is conducted. (A, B) The highlighted areas indicate time-frequency epochs with significant modulation, determined using a cluster-based permutation test (CTD: $p < 0.05$, Wilcoxon test). (A–C) Scalp topographies display oscillatory activity in the theta range. Source estimation for theta activity is provided for Vertex and the TMS effect. Magenta lines on the cortex represent areas where BOLD activity correlates with Pa in the fMRI experiment (see Fig 3). All source results are presented on a high-resolution mesh for visualization purposes only. The data underlying this figure can be found at https://osf.io/zd3g7/. CTD, cluster threshold detection; IPS, intraparietal sulcus; PPC, posterior parietal cortex.

with ambiguous probabilities in the prefrontal cortex. Since there is no learning in our experimental task, we cannot rule out whether the latter activity is just a surprise signal related to expectation violation or has a role in value updating and learning. Nevertheless, building an expectation incorporating environmental ambiguity is necessary in both cases. The interruption of value signals during the decision-making period supports this interpretation. Given

these compelling findings, we posit the possibility of bidirectional communication between the frontal and parietal regions. Our study's outcomes collectively suggest that the parietal cortex is causative in this cognitive process. Although we did not find evidence of differential computational processing between the 2 parietal areas stimulated, it is possible that limitations in the experimental design may have hindered the detection of more subtle differences in the uncertainty computation flow. Notably, overall the behavioral adjustments induced by TMS may be interpreted as strategic adaptations without altering the overall accumulation of rewards [40]. Further research is necessary to understand better the exact nature and mechanisms of the involvement of partial computation in the learning process.

Our results suggest a specific computation occurring in the parietal cortex when humans face ambiguous situations, although other research may reveal alternative interpretations. Under ambiguity, the parietal activation can also be interpreted as sensing the necessity to reduce uncertainty throughout learning, valuing, and categorizing. Thus, interrupting parietal activity could impair categorization processing, generating more straightforward decision rules. Recent research in mice has shown a causal role for the parietal cortex in new but not well-learned sensory stimuli categorization [41]. The parietal cortex takes part in learning and categorization processes before new stimuli have been incorporated into existing categories [41]. The correlation between parietal cortex activity and the degree of ambiguity in the decision revealed by our findings might be associated with a process to reduce the uncertainty that an ambiguous stimulus evokes. Thus, parietal activity may play a role in using previous knowledge and experience in categorical choices [19,41]. Thus, the effect of parietal TMS perturbation could be interpreted as using a simpler heuristic with less categorization processing. For example, the subject simply chooses the option based on the associated reward. Heuristics are crucial in complex situations because they are simplified decision rules that help individuals deal with problems requiring high cognitive investment [42,43]. Compared with younger adults, older adults show different parietal activity when faced with a decision under uncertain probability and different heuristic decision-making [44]. Comparative studies indicate that nonhuman primates, as do adult humans, show ambiguity aversion, revealing that this situation entails a high cognitive cost [2,3]. Interestingly, most primate species use simple heuristics to face ambiguous decisions, and only great apes consider the ambiguity of the information in the decision process [42]. In accordance, researchers in developmental neuroscience have shown that children do not show ambiguity aversion [45] as adults and adolescents do [46]. A recent study shows that the IPS implements complex heuristics in sequential decision-making tasks [47]. In light of our current results, the interference of the parietal cortex can be interpreted here as preventing the use of more complex computation for managing ambiguity.

Considering the preceding evidence, the correlation between parietal activity and the degree of uncertainty can reflect a high cognitive demand. Indeed, previous findings have shown that regardless of the sensory properties of the stimuli, parietal activity has been largely related to numeric magnitude [48,49]. The IPS has been associated with several numerical and spatial operations in humans, including arithmetic calculations and spatial rotation [50]. Thus, the IPS appears to be a highly sensitive region for the processing and manipulation of magnitudes across various dimensions, including abstract numbers, space, and time [50,51]. In our results, the parietal interference operates on a specific parameter in the computational model related to incorporating the uncertainty of ambiguous information. Although such an operation might involve a high cognitive demand, the parietal activity was independent of other proxies of difficulty, such as reaction time. We also found no evidence that parietal stimulation generated a laterality bias in choice. The preceding is important since the right IPS has been related to spatial attention [32]. According to fMRI research, parietal activity during decision-making under uncertainty is not influenced by general attentional load [52]. In this context,

parietal activity seems to be better understood as a specific computation of uncertainty rather than as a general cognitive load.

During outcome monitoring, we report frontal oscillatory activity related to expectation violation based on sensing less uncertainty in the decision under ambiguity. Prefrontal oscillatory activity has a widely studied role in cognitive control and working memory [25,53–55]. Extensive research has shown that prefrontal delta and theta activities correlate with prediction errors [18,21–23,56–59]. Prior evidence showed that parietal and frontal areas sustain effective connectivity during and after a decision, and such activity biases follow decision-making [19,20]. Research using the EEG technique has demonstrated that frontal oscillatory activity correlates with the uncertainty and unexpectedness of an event [21,22] and that it is associated with future exploration strategies [21]. According to prior research, the source of this oscillatory activity is in the MCC [23,60]. It has been hypothesized that MCC contains multiple circuits participating in sensing diverse and relevant internal and environmental variables, such as information about reward, punishment, and uncertainty [61]. Moreover, this area is part of a network involving basal ganglia that presents activity related to outcome uncertainty [62]. Specifically, this network has been proposed as regulating behavior to obtain uncertainty-resolving information [62]. Studies have found that when stimuli have information important to resolving uncertainty, it affects visual search behavior [63,64]. Interestingly, both MCC and the parietal cortex have been linked to such behavior [13,64]. Thus, the IPS—MCC connectivity generated by the ambiguous information could be interpreted as a mechanism to contrast and update the uncertainty of the chosen event. Notably, intolerance to uncertainty has largely been related to anxiety, a pervasive symptom in a broad spectrum of psychiatry and neurological diseases [65]. Accordingly, EEG frontal oscillatory activity reflecting MCC activity is moderated by anxiety and predicts adaptive behavioral adjustments under uncertainty [66]. Furthermore, we have observed an oscillatory modulation during decision-making in a region proximate to the FEF. The FEF is interconnected with parietal areas receiving visual inputs and encoding various features pertinent to decision-making, functioning as an evidence accumulator [67,68]. In addition to its role in perceptual decision-making, the FEF has been implicated in value-based decision-making [69] and the encoding of past reward history [70]. Therefore, the modulation of FEF oscillations may reflect the perturbation due to the TMS of a prior history of prediction errors when calculating ambiguous probabilities. While prior studies have associated this region with decision-making under conditions of uncertainty [9], its precise functional role under uncertainty remains an area of ongoing investigation.

In summary, we took advantage of the sequential use of fMRI and TMS-EES studies to localize and interfere with model-derived signals related to the use of ambiguous probabilities to provide causation. Our results demonstrate a causal implication of the parietal cortex in managing ambiguity during decision-making and assigning uncertainty during decision-making processing. Additionally, we tested whether the localized perturbation in the parietal cortex spreads through the cortex and alters neural processing in remote areas. Specifically, we demonstrated a decrease in the signals related to the value of the chosen option during decisions and related to violation expectation in the MCC once participants evaluated the outcome of their decisions. As a result, the evidence provided here contributes to generating deep insight into the cognitive and neural mechanisms underlying decision-making in situations of ambiguity. Notably, difficulties dealing with uncertainty or ambiguity commonly result in anxiety [65]. Hence, the mechanism we identified here could become a potential target for further studies in several neuropsychiatric symptoms that have been associated with the perception and the computation of uncertainty, such as those present in autism spectrum [71] and obsessive-compulsive disorders [72].

## Materials and methods

### Participants

Seventy-four healthy participants between the ages of 18 and 45 participated in the experimental protocol approved by the Ethics Committee of the Universidad del Desarrollo, Chile (Folio 2020–67). Thirty-nine participants took part in the fMRI session, 24 participated in the EEG-TMS session (9 of them also participated in the fMRI session), and 20 in a behavioral replication sample. All had normal or corrected to normal vision, no color vision impairment, no history of neurological disease, and no current psychiatric diagnosis or psychotropic prescriptions. All participants gave their written informed consent. Experiments were conducted in the Social Neuroscience and Neuromodulation Laboratory at the Centro de Investigación en Complejidad Social (neuroCICS) at the Universidad del Desarrollo, the Unidad de Imágenes Cuantitativas Avanzadas (UNICA) at the Clínica Alemana de Santiago, and Grenoble Institut Neurosciences at the Univ. Grenoble Alpes, Inserm, U1216.

### Task

All participants completed the probabilistic decision-making (PDM) task [73] in which they had to choose between 2 probabilistic options with rewards. Each option was represented by a bar color (on each side of the screen) and associated with a probability of being selected, represented by the length of a colored bar placed in the center of the screen, and a reward, represented by a number placed above each colored bar. These numbers represent real monetary incentives (see below). The options had random, complementary probabilities and rewards, with the option having the highest visible bar (highest probability) having the lowest reward and vice versa. After the participant had made a selection (approximately 2 to 6 s), the rewarded option was indicated with either a green circle if participants chose the rewarded option or with a red circle otherwise. Feedback presentation (red or green circle) lasted for 3 s. If the participant chose the rewarded option, they received the associated reward, otherwise, they received no money. Participants completed this task under 2 conditions: no-ambiguity and ambiguity. In the former condition, participants saw the full extension of the color bar, with complete information related to the probability distribution of possible outcomes (i.e., risk or first-order uncertainty). In the latter condition, a gray mask partially hid part of the extension of both bars. The size of this mask could vary from 40% to 80%. In these cases, participants had incomplete information about the probability distribution of possible outcomes (i.e., ambiguity or second-order uncertainty). The task was programmed and presented using Presentation Software (Neurobehavioral Systems TM).

In the fMRI experimental session, participants completed 40 trials: 20 for the no-ambiguity condition and 20 for the ambiguity condition in 5-trial blocks. In the TMS-EEG experimental session, participants completed 240 trials in 10-trial blocks per condition (no-ambiguity and ambiguity). Each participant completed 6 runs of TMS stimulations, consisting of 2 runs of 40 trials with TMS interference on the PPC (MNI x = 14, y = −64, z = 56), 2 runs of 40 trials with TMS interference at the IPS (MNI x = 46, y = −44, z = 57), and 2 runs of 40 trials with TMS interference at the vertex, as an active control condition. The order of these 6 runs was randomly selected for each participant. Stimulation was applied 200 and 300 ms before the Feedback epoch with a double inhibitory pulse separated by 100 ms. The TMS target regions were calculated based on group analyses of the fMRI session. For the behavioral replication sample, the options were presented as 2 independent lotteries. During the ambiguity condition, the areas behind the mask were not revealed in order to explore for possible bias. Each participant completed 150 trials in 5-trial blocks per condition (no-ambiguity and ambiguity), with a break every 50 trials.

## Statistical analysis of the behavior

The participants' answers were analyzed with a computational cognitive approach. All computational cognitive models were fitted using prospect theory, which assumes that the following equation defines the expected subjective value $U_l$ of an option (indicates left option) that individuals use to make a decision.

$$U_l = v(x_l)\pi(P_{all_l}) - v(x_r)\pi(P_{all_r}) \tag{4}$$

In Eq 4, subindices "l" and "r" represent the left and right options, respectively. $v(.)$ represents the value function, $x_l$ and $x_r$ denote the potential outcome of each option associated with the left or right option, respectively. We used the following equation,

$$v(x) = x^\alpha, \tag{5}$$

where $\alpha$ determines the concavity of the value function. $P_{all}$ is the probability of a gain, whereas $\pi(.)$ are the subjective decision weights assigned to these probabilities. To accommodate for the existence of unknown probabilities (i.e., for ambiguity condition), the probability $P_{all}$ by which the outcome x occurs is defined by the Eqs 1, 2, 3, and 6 (see Results).

$$\pi(P_{all}) = \frac{P_{all}^y}{\left(P_{all}^y + (1 - P_{all})^y\right)^{1/y}} \tag{6}$$

The extent by which the ambiguity area $P_a$ is assigned to each option is modulated by 2 parameters: $\tau_i$ that represents the ratio of $P_a$ effectively assigned, and $\tau_b$ that represents the ratio by which the subject biases one of the 2 options. The models where $\tau_b$ was set to 0.5 involved a process of unbiased (homogeneous) assignment between options (left or right). Additionally, we explored an alternative bias parameter $W_r$ that influenced the reward estimation ($\alpha$ parameter of Prospect Theory, Eq 5) under ambiguity given by the following equation:

$$v(x_l) = \begin{cases} x_l^{(\alpha + W_r)} & x_l \geq x_r \\ x_l^{(\alpha)} & x_l \prec x_r \end{cases} \tag{7}$$

The probability of choosing the left option for a given subjective value is computed using a logistic choice rule, wherein $\beta_1$ is an inverse temperature parameter representing the degree of stochasticity in the choice process and $\beta_0$ is a bias parameter.

$$\theta(U_l) = \frac{1}{1 + e^{-\beta_l(U_l - \beta_0)}} \tag{8}$$

All parameters were estimated using a hierarchical Bayesian approach that uses the aggregated information from the entire population sample to inform and constrain the parameter estimates for each individual. The hierarchical structure contains 2 levels of random variation: the trial (i) and participant (s) levels. At the trial level, choices were modeled following a Bernoulli process:

$$y(s, i) \sim bern(\theta(U_l - U_r)). \tag{9}$$

At the participant level, the model parameters were constrained by group-level hyperparameters.

The parameters $\tau_b$ were restricted to be between 0 and 1 using a Beta distribution.

$$\tau_b(s) \sim beta(\mu_{\tau b} \kappa_{\tau b}, (1 - \mu_{\tau b})\kappa_{\tau b}). \tag{10}$$

Where $\mu_\tau$ represents the mean and $\kappa_\tau$ represents the dispersion of the beta distribution.

The parameters $\beta$ and $W_r$ at the participant level were parameterized using normal distributions. The $\tau_i$, $\gamma$ and $\alpha$ parameters at the participant level were also parameterized using normal distributions and restricted to positive values.

$$\alpha_{(s)} \sim normal(\mu_\alpha, \sigma_\alpha) \tag{11}$$

$$\beta_{(s)} \sim normal(\mu_\beta, \sigma_\beta) \tag{12}$$

$$\gamma_{(s)} \sim normal(\mu_\gamma, \sigma_\gamma). \tag{13}$$

We assumed flat distributions for each parameter at the highest level of the hierarchy (hyperparameters).

$$\mu_{(\alpha, \tau i, \gamma)} \sim unif(0.01, 100) \tag{14}$$

$$\mu_{(Wr, \beta)} \sim normal(0, 100) \tag{15}$$

$$\sigma \sim unif(0.001, 100) \tag{16}$$

$$\mu_{(\tau b)} \sim beta(1, 1) \tag{17}$$

$$\kappa_{\tau b} \sim unif(0.001, 100). \tag{18}$$

The posterior inference of the parameters in the hierarchical Bayesian models was performed via the Gibbs sampler using the MCMC technique, which was implemented in JAGS using R software. A minimum of 10,000 samples were drawn from an initial burn-in sequence. Subsequently, a total of 10,000 new samples were drawn using 3 chains, each of which was derived based on a different random number generator engine using different seeds. We increased the length of the burn-in sequence if the chains did not meet the criteria for convergence, as outlined below. We applied a thinning of 10 to this sample, resulting in a final set of 3,000 samples for each parameter. This thinning was used to avoid autocorrelation among the final samples for the parameters of interest. We conducted Gelman–Rubin tests for each parameter to confirm the convergence of the chains. All latent variables in our models had a Gelman–Rubin statistic near 1, which suggests that all 3 chains converged to the target posterior distribution.

Additionally, the behavior was also analyzed using mixed-effect logistic regression, assuming no specific ambiguity computation.

$$Left \sim P_{objl} + R_{wl} + P_a + P_{objl} : P_a + P_{objl} + P_a. \tag{19}$$

The full interaction logistic model was tested but presented higher AICs, indicating worse adjustments.

## Anatomical data

All participants underwent a 3D anatomical MPRAGE T1-weighted and T2-weighted magnetic resonance imaging scan on a 3T Siemens Skyra (Siemens AG, Erlangen, Germany) no more than 3 months before the TMS-EEG sessions or together with the fMRI sessions. The anatomical volume consisted of 160 sagittal slices of an isotropic voxel ($1 \times 1 \times 1$ mm), covering the whole brain. The scalp and cortical surfaces were extracted from the T1-weighted/T2-weighted corrected anatomical MRI using a pipeline available from the Human

Connectome Project. Thus, a surface triangulation was obtained for each envelope [74]. The individual high-resolution cortical surfaces (approximately 300,000 vertices per cortical surface) were down-sampled to approximately 8,000 vertices. Additionally, a five-layer segmentation based on T1-weighted and T2-weighted was carried out using the algorithm implemented by the SimNIBS tool and SMP12. The cortical mesh and five-layer segmentation served as image supports for the EEG source estimation (see below).

## Functional MRI data

For the functional images, volumes of the entire weighted echo-planner T2* brain were acquired while the experimental task was executed ($3 \times 3 \times 3$ mm voxels). Participant volumes were coregistered to 2-mm standard imaging using the nonlinear algorithm implemented in FSL. The BOLD signal was analyzed using different models, including motion correction parameters (MC). During decision-making periods, we fitted 2 models as follows.

fMRI Model 1:

$$\text{BOLD} \sim P_{all[A]} + P_{all[nA]} + P_a + R_w + RT + Am + MC \tag{20}$$

fMRI Model 2:

$$\text{BOLD} \sim P_{all(\tau i=0)} + P_{all(\tau i=1)} + P_a + R_w + RT + Am + MC. \tag{21}$$

Am is a dummy regressor capturing the "state" or baseline activity that the participants had in the ambiguity condition, and the RT is a reaction-time regressor as a proxy of difficulty. For model 1, $P_{all}$ for both conditions (Ambiguity [A] and No-ambiguity [nA]) was orthogonalized to $P_a$ in order to obtain unique activity related to probability computation independent of the degree of ambiguity. Additionally, in fMRI-Model 2 $P_{all(\tau i = 0)}$ and $P_{all(\tau i = 1)}$ were orthogonalized with each other in order to obtain their independent contribution to the signal during the ambiguity condition.

For the BOLD signal during outcome evaluation (feedback), we used the following regressors of interest: Win (a dummy regressor indicating that the chosen option was rewarded), $R_w$ (the amount of the obtained reward), uPE-$P_{all}$[A] (the unsigned prediction error of the fact to win or not to win given by the $P_{all}$ of the chosen option in Ambiguity conditions), uPE-$P_{all}$[nA] (the unsigned prediction error of the fact to win or not to win given by the $P_{all}$ of the chosen option in no-ambiguity conditions), uPE-$R_w$ (the unsigned prediction error of the amount of the obtained reward). All regressors were convolved using a double gamma function.

## EEG recordings

We used TMS-compatible EEG equipment (BrainAmp 64 DC, BrainProducts, http://www.brainproducts.com/). EEG was continuously acquired from 64 channels (plus an acquisition reference (FCz) and a ground). TMS-compatible sintered Ag/AgCl-pin electrodes were used. The signal was band-pass filtered at DC to 1,000 Hz and digitized at a sampling rate of 5,000 Hz. Skin/electrode impedance was maintained below 5 $k\Omega$. Electrode impedances were re-tested during pauses to ensure stable values throughout the experiment. The positions of the EEG electrodes were estimated using the neuronavigation system used for the TMS.

## EEG-TMS protocol

TMS was applied during task performance and during EEG recordings. Participants were instructed to maintain central fixation and minimize eye blinks and other movements during the recording blocks. Double biphasic TMS pulses were delivered over the right IPS (TMS_ips,

MNI [46, −44, 57]), the right PPC (TMS$_{ppc}$, MNI [14, −64, 56]), and the Vertex (TMS$_{vertex}$, MNI [0, −29, 77]; see Results, Figs 3 and 4A) using a 70 mm figure-of-eight TMS coil connected to Mag and More Stimulator. A neuronavigation system was used to identify individual stimulation points (individual structural MR scans, native space) in the nearest gray matter areas to the no-linear inverse co-registration of the individual anatomy (FSL algorithm, default parameters). TMS coil positioning and orientation with regards to brain x, y, and z axes (*yaw*, *pitch*, and *roll*) were optimized so that the electric field impacted perpendicular to the target region, maximizing the induced current strength [75,76]. This approach results for all subjects with approximately an angulation in a horizontal plane (yaw) with regards to the interhemispheric fissure of 45° for the IPS and 0° for the PPC and the vertex. For each trial and for both tasks, 2 consecutive single TMS pulses were delivered before the feedback presentation (−300 and −200 ms pre-stimulus onset) with an interpulse interval of 100 ms in order to interfere with target activity with a 100 to 200 ms window that has been used in prior work [77,78] and has been demonstrated to inhibit motor potential [79]. TMS intensity was fixed at 120% of the individual resting motor threshold (TMS intensity ranging from 54% to 78% of the maximum machine power and a mean of 63%). Each TMS session included 6 runs. In each run, 40 two-pulse TMS bursts were delivered trial by trial, leading to 80 pulses per run over a block duration of about 11 min. Pauses for a minimum of 5 min duration separated each run. Each TMS-EEG experiment thus contained a total of 480 active TMS pulses (including those delivered at the vertex). The rationale for this block design was to obtain a balance that maximizes the number of trials per condition while maintaining a single TMS-EEG session for each subject [80,81]. This approach allowed us to enhance our statistical power through within-subject analysis. Two 5-min EEG resting-state recordings were performed before and after the 6 blocks. The duration of the experiment was around 180 min: 1 h for setting the EEG electrodes at stable and adequate impedances, 1 and half hours of recordings, and 30 min for the electrode MRI localization and experiment finalization. The TMS protocol respected at all times past and current safety recommendations regarding stimulation parameters (intensity, number of pulses, and ethical requirements [82–84]).

## EEG preprocessing and TMS artifact removal

Preprocessing was performed in multiple steps. We first detected the slow decay component of the TMS artifact. To this end, we segmented 1-s windows containing TMS pulses, automatically detected a period starting 10 ms pre to 20 ms post to the respective TMS peak and removed this from the signal. We applied an independent component analysis (ICA) to this signal using the Runica algorithm provided by the EEGLAB toolbox (https://sccn.ucsd.edu/eeglab/). Thus, we looked for a stereotype component with local bipolar distribution over the TMS site pulse. In the second step, we segmented the raw signal in the time widows of analysis (−1.5 s to 2 s after feedback onset). Then, we removed the segment between −10 to 30 ms around the TMS peak and replaced it with an inverse-distance weighted interpolation [Y = sum(X/D^3)/sum(1/D^3)] plus a Gaussian noise with the standard deviation extracted to a reference period set to be 55 to −15 ms before the respective first TMS peak of the double pulse and 0 of the mean. Then, we removed the TMS ICA components computed in the first step. This procedure effectively removed the direct (non-physiological) and other TMS artifacts (e.g., TMS-locked artifacts at electrodes directly in contact with the TMS coil) without introducing discontinuities, important for the later time-frequency analysis [75,85]. Following these steps, we down-sampled EEG data to 1,000 Hz and used a preprocessing pipeline developed for prior work [54,55,86–88]. The EEG data was 0.1 to 45 Hz band-pass filtered. Artifacts were first automatically detected using a threshold of 150 $\mu V$ and a power spectrum greater

than 2 std. dev. for more than 10% of the frequency spectrum (0.5 to 40 Hz). Blinking was extracted from the signal by means of ICA. The remaining trials that included artifacts detected by visual inspection of the signal were eliminated. The mean of artifact-free trials was 229 out of 240, ranges: [182 240]. Finally, the signal was re-referenced offline to the average of all electrodes for the subsequent analyses.

Time-frequency (TF) distributions were obtained by means of the wavelet transform in a time window between −1.5 and 2 s around feedback onset. To this end, the signal x(t) was convolved with a complex Morlet's wavelet function. Wavelets were normalized, and the width of each wavelet function was chosen to be 5 cycles. Thus, we obtained the phase and amplitude per each temporal bin (in steps of 10 ms) and frequency (from 1 to 40 Hz in steps of 1 Hz). For all power spectrum analyses, we used the dB of power related to a baseline during the fixation phase (at the beginning of the experiments). To avoid edge artifacts, only the period between −0.25 to 0.8 s over the segmented signals was used for additional analyses.

We calculated general linear models for each subject based on single-trial wavelet transform (first-level analysis). We used the following regressor for this analysis: Win (dummy regressor), $R_w$, uPE-$P_{all(ri = 0)}$, uPE-$P_{all(ri = 1)}$, uPE-$R_w$, Am (dummy regressor), and a regressor for each TMS stimulation ($TMS_{ips}$, $TMS_{ppc}$) and the interaction between the TMS regressor with the preceding regressors. Thus, per each regressor and subject, we obtained a 3D matrix (time, frequency, electrode), which we used in the second-level analysis. For the analyses of the frontal electrodes of interest, the 3D matrix, including only the selected ones (Fz, CFz, Cz, F1, FC1, C1, F2, FC2, C2), was averaged in the electrode dimension. We explored consistent modulations in the same condition. For this, we used the Wilcoxon signed sum test, evaluating whether the mean is different from zero. All comparisons were corrected for multiple comparisons using a cluster-based permutation test (see below) [89] or false discovery rate (FDR) for a priori selection of a frequency-time window of interest in source analysis.

## Cluster-based permutation test

In order to correct for multiple comparisons in the time-frequency analysis, we carried out a permutation test [89]. Here, clusters of significant areas were defined by pooling neighboring sites that showed the same effect ($p < 0.05$ in the statistical test carried out in sites of either the time-frequency chart or the sources, e.g., Wilcoxon test). The cluster-level statistics were computed as the sum of the statistics of all sites within the corresponding cluster. We evaluated the cluster-level significance under the permutation distribution of the cluster that had the largest cluster-level statistics. The permutation distribution was obtained by randomly permuting the original data. Specifically, for each subject, we carried out null models, wherein the same structure of the original model was preserved, but the regressor tested was permuted. After each permutation, the original statistical test was computed (e.g., Wilcoxon), and the cluster-level statistics of the largest resulting cluster were used for the permutation distribution. After 5,000 permutations, the cluster-level significance of each observed cluster was estimated as the proportion of elements of the permutation distribution greater than the cluster-level statistics of the corresponding cluster.

## EEG source estimation

The neural current density time series at each brain location were estimated by applying a minimum norm estimate inverse solution LORETA algorithm with unconstrained dipole orientations in single-trial signal per condition and subject, implemented in Brainstorm. A tessellated cortical mesh for individual anatomy was used as a brain model to estimate the current source distribution. We defined approximately $3 \times 8,000$ sources constrained to the segmented

cortical surface (3 orthogonal sources at each spatial location). We computed a five-layer continuous Galerkin finite element conductivity model (FEM), as implemented in DUneuro software [90], and the physical forward model. To estimate cortical activity at the cortical sources, the recorded raw EEG time series at the electrodes were multiplied by the inverse operator to yield the estimated source current at the cortical surface as a function of time. Since this is a linear transformation, it does not modify the frequencies of the underlying sources. It is, therefore, possible to undertake time-frequency analysis on the source space directly. In this source space, we first reduced the dipole of each vertex to one, selecting the component with the greater variance using the PCA algorithm. We then computed frequency decomposition using the Wavelets transform. To minimize the possibility of erroneous results, we only present source estimations if there are statistically significant differences at both the electrode and source levels (i.e., differences that survive multiple comparison corrections).

## Supporting information

**S1 Fig. Behavioral Results for replication sample.** (**A**) Rate of choice where individuals preferred the highest probability per conditions. (**B**) Rate of choice where subjects preferred the highest probability per degree of ambiguity. Black dots represent the rate per individual. Color rectangles indicate the number of individual decisions; red represents the maximum, and light gray, represents the minimum account. The blue line represents the linear regression, and the gray area is the standard error. (**C**) Model fitting comparison using DIC, red, and LOOIC, green. (**D**) Posterior distribution of the key parameters for each model. Black dots represent the mean of the distribution, and black lines the 95% high-density intervals. The colored areas represent the complete posterior distribution. (E) Correlation between decision shift (difference between the rate of choices that subject prefers the highest probability between condition, Ambiguity less Non-ambiguity) and parameters. Red dots represent each subject. The solid blue line represents the linear regression, the dotted blue line the LOESS regression, and the gray area represents the standard error; * indicates $p < 0.05$, ** $p < 0.01$, *** $p < 0.001$. The data underlying this figure can be found at https://osf.io/zd3g7/.
(PNG)

**S2 Fig. Parameters recovery.** Posterior distribution of $\tau_i$ parameter recovery from simulated dated generated by different tau parameters. For the simulation, all the other model parameters were fixed using the mean of the posterior distribution fitted from the real data. The data underlying this figure can be found at https://osf.io/zd3g7/.
(PNG)

**S3 Fig. Distribution of model variables derived from simulations as a function of the $\tau_i$ parameter.** (A) Mean reward obtained in a game as a function of the $\tau_i$ parameter. (B) Mean rate of positive feedback or winning obtained in a game as a function of the $\tau_i$ parameter. (C) Mean probability prediction error (PPE) obtained in a game as a function of the $\tau_i$ parameter. (D) Mean reward prediction error (RPE) obtained in a game as a function of the $\tau_i$ parameter. (E) Mean rate of choosing the more probable option in a game as a function of the $\tau_i$ parameter. The data underlying this figure can be found at https://osf.io/zd3g7/.
(PNG)

**S1 Table. Behavioral models.** Parameters of the different behavioral model adjusted. The data underlying this table can be found at https://osf.io/zd3g7/.
(PDF)

**S2 Table. fMRI models.** Significant cluster for the fMRI models. The data underlying this table can be found at https://osf.io/zd3g7/.
(PDF)

**S3 Table. Behavioral TMS models.** Parameters of the different behavioral model adjusted for TMS experiments. The data underlying this table can be found at https://osf.io/zd3g7/.
(PDF)

# Acknowledgments

We want to thank Leonie Kausel and Anne Bliss for proofreading the manuscript.

# Author Contributions

**Conceptualization:** Gabriela Valdebenito-Oyarzo, Antoni Valero-Cabré, Rafael Polania, Pablo Billeke.

**Formal analysis:** Gabriela Valdebenito-Oyarzo, Rafael Polania, Pablo Billeke.

**Funding acquisition:** Julien Bastin, Pablo Billeke.

**Investigation:** Gabriela Valdebenito-Oyarzo, María Paz Martínez-Molina, Alejandra Figueroa-Vargas, Pablo Billeke.

**Methodology:** Gabriela Valdebenito-Oyarzo, Julien Bastin, Rafael Polania, Pablo Billeke.

**Project administration:** Gabriela Valdebenito-Oyarzo, Alejandra Figueroa-Vargas, Pablo Billeke.

**Resources:** Francisco Zamorano, Josefina Larraín-Valenzuela, Ximena Stecher, César Salinas.

**Writing – original draft:** Gabriela Valdebenito-Oyarzo, Patricia Soto-Icaza, Antoni Valero-Cabré, Rafael Polania, Pablo Billeke.

**Writing – review & editing:** Gabriela Valdebenito-Oyarzo, Patricia Soto-Icaza, Antoni Valero-Cabré, Rafael Polania, Pablo Billeke.

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
