## [Editor Report · Decision Letter 0]

2 May 2023

Dear Dr Billeke, 

Thank you for submitting your manuscript entitled "A causal role for the parietal cortex in ambiguity computations in humans" for consideration as a Research Article by PLOS Biology.

Your manuscript has now been evaluated by the PLOS Biology editorial staff as well as by an academic editor with relevant expertise and I am writing to let you know that we would like to send your submission out for external peer review.

Once your full submission is complete, your paper will undergo a series of checks in preparation for peer review. After your manuscript has passed the checks it will be sent out for review. To provide the metadata for your submission, please Login to Editorial Manager (https://www.editorialmanager.com/pbiology) within two working days, i.e. by May 04 2023 11:59PM.

Kind regards,

Luke

Lucas Smith, Ph.D.

Associate Editor

PLOS Biology

lsmith@plos.org

---

## [Decision Letter · Decision Letter 1]

28 Jul 2023

Dear Dr Billeke,

Thank you again for your patience while your manuscript "A causal role for the parietal cortex in ambiguity computations in humans" was peer-reviewed at PLOS Biology. Your study has now been evaluated by the PLOS Biology editors, an Academic Editor with relevant expertise, and by several independent reviewers. 

In light of the reviews, which you will find at the end of this email, we would like to invite you to revise the work to thoroughly address the reviewers' reports.

As you will see below, the reviewers report that the study is generally well done and that the findings are of interest to the field. However each reviewer has raised a number of important points and suggestions aimed at strengthening the study and improving the presentation, and we think these will need to be carefully addressed before we can consider your study further at PLOS Biology. 

Given the extent of revision needed, we cannot make a decision about publication until we have seen the revised manuscript and your response to the reviewers' comments. Your revised manuscript is likely to be sent for further evaluation by all or a subset of the reviewers.

**IMPORTANT - SUBMITTING YOUR REVISION**

*Re-submission Checklist*

*Published Peer Review*

*PLOS Data Policy*

*Blot and Gel Data Policy*

Sincerely,

Luke

Lucas Smith, Ph.D.

Senior Editor

PLOS Biology

lsmith@plos.org

REVIEWS:

Reviewer #1: In this work, the authors demonstrated that the parietal cortex plays a causal role in computing ambiguous probabilities, using a very nice design that embrace a robust and multi-method perspective (behavioural modeling, fMRI, TMS, EEG). They found that participants elevated the uncertainty associated to their decisions and that parietal activity correlated with the underestimation of the uncertainty during decision-making. Moreover, disruption of parietal activity by means of TMS influences the assignment of ambiguous probabilities, increased the uncertainty associated to decision-making. In addition, they found that the midcingulate cortex encodes prediction errors and increases its connectivity with the parietal cortex during outcome processing. 

The work is well done and provides an exciting contribution to the topic. In addition, they followed a sequential approach in which the results of one neuroscientific technique (e.g., fMRI) guided the investigation that was carried out with another set of techniques (TMS-EEG), enhancing the validity of the approach and robustness of the findings. 

In what follows I provide some comment which I hope can support the authors in further improving this already excellent work. 

Comments:

1) In the results section, you stated that "we first aimed to examine the intra-block cumulative effect of trial-by-trial TMS on behavior. To do this, we analyzed the behavioral effects in the last 20 trials of each 40-trial block of the same TMS stimulation." 

I feel that it might be helpful to further clarify why you expect a cumulative effects of the TMS protocol applied.

2) Connected to the previous point, you reported in the text that "in the TMS-EEG experimental session, participants completed 6 runs of TMS stimulations, consisting of 2 runs of 40 trials with TMS interference on the PPC, two runs of 40 trials with TMS interference at the IPS, and two runs of 40 trials with TMS interference at the vertex, as an active control condition." Considering that you have analyzed the behavioral effects in the last 20 trials of each 40-trial block of the same TMS stimulation, it means that, you have considered the contribution of 40 trials, in turn divided in non-ambiguous and ambiguous trials, for each TMS condition? I would like to have more clarification on this, considering that this number might be too small to have a stable measure of the effect of TMS on behavioral indices.

3) If I understand correctly, the activity of the right parietal regions is correlated with a tendency to calculate options as more certain than they actually are, and the disruption of their activity by TMS leads participants to change their decisional strategy, behaving as if they perceive greater uncertainty in decision-making. These findings resemble recent research (e.g., Tarasi et al., Prog. Neurobiology 2022) that showed that right parietal areas are involved in the modulation of decision-making strategies. This same research, however, demonstrates that this strategical adjustment is not linked to a concurrent increase in objective performance. Would it be possible to demonstrate whether TMS stimulation of the parietal area may have influenced the participants' objective performance or whether it modulated only the decision-making strategy? For example, is it possible to show whether the "final" reward obtained by the participants differs significantly according to the TMS condition or remains stable between the tested conditions?

4) In the paper, the activity from the parietal to the frontal is mainly treated since the former would provide outcome predictions made through ambiguity computation needed to calculate the prediction error. Is it possible to discuss whether there could also be a flow that travels in the opposite direction (from frontal-to-parietal) that "adjusts" the ambiguity computation after a commission of an error?

5) Would it be possible to demonstrate whether there are any oscillatory signatures of the role of the parietal regions when computing ambiguous probabilities?

Reviewer #2: The behavioural results that the authors demonstrated are convincing. The shift in behavior during ambiguous decisions is associated with a selective decrease in probability weighting.

The model that the authors apply is appropriate. It is great that the robustness of the model was confirmed by an independent data set.

fMRI experiments and analyses were carefully conducted and the reviewer was convinced by the authors' conclusion that activity in the bilateral IPS and PPC reflects the degree of ambiguity.

1) Causal evidence by the TMS experiment is also convincing but it is a bit uncertain for the reviewer about the difference in causal impact by stimulation between IPS and PPC.

The authors state that they merged data from IPS and PPC stimulation as there are no differences in their effects on behavior. But the reviewer thinks that it is worth demonstrating the data with IPS stimulation and PPC stimulation separately.

2) Especially it is interesting if the authors show EEG data with TMS stimulation separately for the IPS stimulation condition and PPC stimulation condition. How was the effect of EEG signals around PPC caused by IPS stimulation? How about EEG signal changes around IPS caused by PPC stimulation? It will help the authors to dissociate the roles of IPS and PPC for value-based decision making with consideration of ambiguity.

3) The authors discuss that correlation between frontal delta and theta activity and prediction errors in uncertain situations is known in previous studies. In the present study they clearly demonstrate a reduction in delta activity in the MCC during feedback caused by parietal disruption. That is one of the most remarkable findings of the study. But the reviewer supposes that there is a possibility that MCC plays a causally essential role in the computation of ambiguous information rather than the parietal cortex. The reviewer proposes that the authors should demonstrate the predictability of behavioural performance by the BOLD signal in PPC, IPS and MCC, separately, during the fMRI experiment and compare them. This analysis will strengthen the authors' point that parietal cortex plays essential role rather than MCC.

4) Please double-check figure legend. Some information, especially statistical values are invisible.

Overall, the manuscript seems to be persuasive and well-organised. The reviewer thinks that it reaches the criterion to publish from PLOS Biology if they sort out the aforementioned issue.

Reviewer #3: The manuscript investigates the role of the parietal cortex in decision-making involving ambiguity, combing computational modeling, fMRI and TMS-EEG technique. In particular, the paper proposes an internal process by which ambiguity is converted into a subjective probability distribution in service of decision-making and tested this proposal using neuroscience tools. Their results point to a role of the parietal cortex which seems to be related to ambiguity processing in a number of different ways. Overall, the research question is important and the paper addresses the question using both correlational and causal tools. However, the writing of the paper should be significantly improved: some writing are quite sloppy, some important technique details are missing, and some figures are not cited in the main text (e.g., Fig 4B). Also, there are a few major questions need to be addressed. 

1. computational model. It would be helpful to include a couple of model simulation to clearly demonstrate the intuition. The paper should provide rationale and discussion of the proposed model, especially in light of the existing literature. Whether and how can the model address well-established findings such as ambiguity aversion? As far as I understand, ambiguity aversion should be closely related to the value of tau_i. But according to the paper, the value estimates of this parameter seem to vary quite a bit across two behavioral datasets, raising questions on the validity of the model. 

Another issue is that the model only considers the decision-making process. In neural data, however, the feedback (and prediction error) was proposed to play an important role. It would be helpful to formally model how feedback would influence subsequent decisions, and provide a quantitative framework for understanding the neural data. 

2. fMRI. I personally find that the writing of the fMRI part was difficult to follow. The paper needs to provide a better explanation of why a specific analysis was performed, and whether there are confounding factors that need to considered. For example, to which extent does P_all ( tau_i = 0) is correlated with P_all ( tau_i = 1)? whether the finding of the left parietal cortex can be explained by the correlation between the two? 

3. TMS. The overall design was quite confusing to me. why focus on the feedback period? Whether and how should feedback influence choices? Whether the trials are related to one another (i.e., what needs to be learned from previous decision)? Do subjects actually show any learning effect in the behavioral experiment? 

For the TMS behavioral results, only mean value and non-parametric p value was reported. The paper needs to include other important statistics such as confidence interval, SEM or quantile, etc.

---

## [Decision Letter · Decision Letter 2]

17 Nov 2023

Dear Dr Billeke,

Thank you for your patience while we considered your revised manuscript "A causal role for the parietal cortex in ambiguity computations in humans" for consideration as a Research Article at PLOS Biology. Your revised study has now been evaluated by the PLOS Biology editors, the Academic Editor and the original reviewers. 

As you will see below the reviewers agree that the revision has largely addressed their original concerns - however Reviewer 2 a lingering request that we think should be addressed in another short revision. We will then assess your revised manuscript and your response to the reviewers' comments with our Academic Editor aiming to avoid further rounds of peer-review, although might need to consult with the reviewers, depending on the nature of the revisions.

As you address the last reviewer comments, please also attend to the following editorial requests: 

1) TITLE: We think the title might flow better if slightly re-arranged. We will ultimately defer to you, but suggest you consider changing it to "The parietal cortex has a causal role in ambiguity computations in humans"

2) ETHICS STATEMENT: Please update the ethics statement in your methods section, to include the approval number for the protocol approved by the Ethics Committee of the Universidad del Desarrollo, Chile. Please also update this to indicate whether consent was written, or not. 

3) DATA: Thank you for providing the underlying data for your study as a deposition to OSF. Can you please add a brief sentence to each relevant figure legend (including supplemental) referencing this data? For example, you can add the sentence "the data underlying this figure can be found at https://osf.io/zd3g7/"

**IMPORTANT - SUBMITTING YOUR REVISION**

*Resubmission Checklist*

*Published Peer Review*

Sincerely,

Luke

Lucas Smith, Ph.D.

Senior Editor

PLOS Biology

lsmith@plos.org

REVIEWS:

Reviewer #1: The authors have adequately addressed all the concerns to full satisfaction. I have no further comments. 

Reviewer #2, Kentaro Miyamoto (note, reviewer 2 has signed this review): The authors fully respond to my comments with additional data analyses. Now, I'm convinced that the causal effects on behaviour by PPC and IPS are comparable. However, we have to reserve that a collection of the larger number of trials for each session is required to draw a firm conclusion about the distinction between the roles of PPC and IPS.

For point 3, it is remarkable that BOLD signal in the parietal ROIs could predict behavioural performance, whereas that in MCC ROI. Direct statistical comparisons of the predictabilities between parietal and MCC ROIs will strengthen the author's point. 

Reviewer #3: The revision has addressed my concerns. I have no further questions.

---

## [Editor Report · Decision Letter 3]

28 Nov 2023

Dear Dr Billeke,

Thank you for the submission of your revised Research Article "The parietal cortex has a causal role in ambiguity computations in humans" for publication in PLOS Biology and thank you for addressing the last editorial and reviewer requests. On behalf of my colleagues and the Academic Editor, Matthew F. S. Rushworth, I am pleased to say that we can in principle accept your manuscript for publication, provided you address any remaining formatting and reporting issues. These will be detailed in an email you should receive within 2-3 business days from our colleagues in the journal operations team; no action is required from you until then. Please note that we will not be able to formally accept your manuscript and schedule it for publication until you have completed any requested changes.

PRESS

Sincerely, 

Lucas Smith, Ph.D.,

Senior Editor

PLOS Biology

lsmith@plos.org